# Node-Level Topological Representation Learning on Point Clouds

## Abstract

Topological Data Analysis (TDA) allows us to extract powerful topological, and higher-order information on the global shape of a data set or point cloud. Tools like Persistent Homology or the Euler Transform give a *single* complex description of the *global structure* of the point cloud. However, common machine learning applications like classification require *point-level* information and features to be available. In this paper, we bridge this gap and propose a novel method to extract node-level topological features from complex point clouds using discrete variants of concepts from algebraic topology and differential geometry. We verify the effectiveness of these topological point features (TOPF) on both synthetic and real-world data and study their robustness under noise.

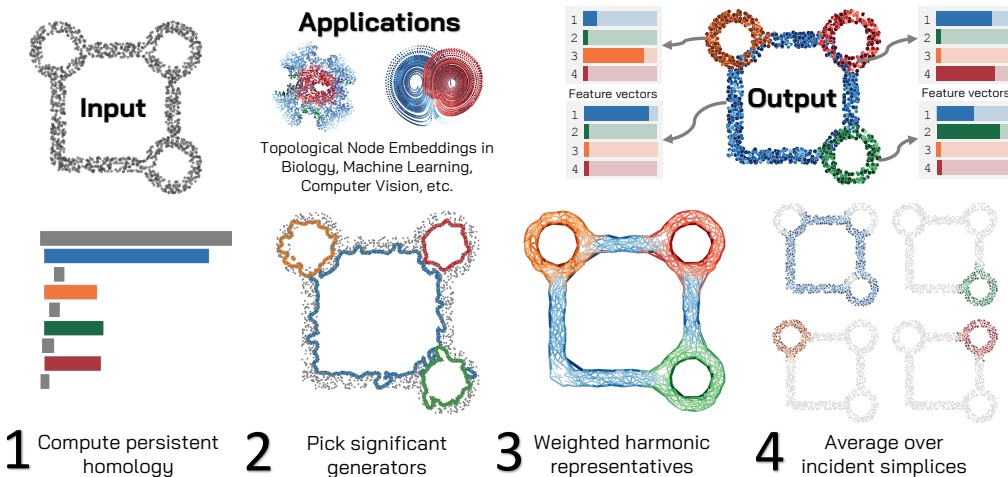

Figure 1: **Schematic of Computing Topological Point Features (TOPF). Input.** A point cloud $X$ in $n$-dimensional space. **Step 1.** To extract global topological information, the persistent homology is computed on an $\alpha$/VR-filtration. The most significant topological features $\mathcal{F}$ across all specified dimensions are selected. **Step 2.** $k$-homology generators associated to all features $f_{i,k} \in \mathcal{F}$ are computed. For every feature, a simplicial complex is built at a step of the filtration where $f_{i,k}$ is alive. **Step 3.** The homology generators are projected to the harmonic space of the simplices. **Step 4.** The vectors are normalised to obtain vectors $\mathbf{e_k^i}$ indexed over the $k$-simplices. For every point $x$ and feature $f \in \mathcal{F}$, we compute the mean of the entries of $\mathbf{e_k^i}$ corresponding to simplices containing $x$. The output is a $|X| \times |\mathcal{F}|$ matrix which can be used for downstream ML tasks. **Optional.** We weigh the simplicial complexes resulting in a topologically more faithful harmonic representative in **Step 3**.

Submitted to 38th Conference on Neural Information Processing Systems (NeurIPS 2024). Do not distribute.

# 1 Introduction

In modern machine learning [39], objects are described by feature vectors within a high-dimensional space. However, the coordinates of a single vector can often only be understood in relation to the entire data set: if the value $x$ is small, average, large, or even an outlier depends on the remaining data. In a 1-dimensional (or low-dimensional) case this issue can be addressed simply by normalising the data points according to the global mean and standard deviation or similar procedures. We can interpret this as the most straight-forward way to construct *local* features informed by the *global* structure of the data set.

In the case where not all data dimensions are equally relevant, or contain correlated and redundant information, we can apply (sparse) PCA to project the data points to a lower dimensional space using information about the *global structure* of the point cloud [51]. For even more complex data, we may first have to learn the encoded structure itself: indeed, a typical assumption underpinning many unsupervised learning methods is the so-called "manifold hypothesis" which posits that real world data can be described well via submanifolds of $n$-dimensional space [36, 21]. Using eigenvectors of some Laplacian, we can then obtain a coordinate system intrinsic to the point cloud (see e.g. [47, 4, 15]). Common to all these above examples is the goal is to construct locally interpretable point-level features that encode *globally meaningful positional information* robust to local perturbations of the data. However, none of these approaches is able to represent higher-order topological information, making point clouds with these kind of structure inaccessible to point-level machine learning algorithms.

Instead of focussing on the interpretation of individual points, topological data analysis (TDA), [9], follows a different approach. TDA extracts a global description of the shape of data, which is typically considered in the form of a high-dimensional point cloud. This is done measuring topological features like persistence homology, which counts the number of generalised "holes" in the point cloud on multiple scales. Due to their flexibility and robustness these global topological features have been shown to contain relevant information in a broad range of application scenarios: In medicine, TDA has provided methods to analyse cancer progression [33]. In biology, persistent homology has been used to analyse knotted protein structures [5], and the spectrum of the Hodge Laplacian has been used for predicting protein behaviour [50].

This success of topological data analysis is a testament to the fact that relevant information is encoded in the global topological structure of point cloud data. Such higher-order topological information is however invisible to standard tools of data analysis like PCA or $k$-means clustering, and can also not be captured by graph models of the point cloud. We are now faced by a situation where **(i)** important parts of the global structure of a complex point cloud can only be described by the language of applied topology, however **(ii)** most standard methods to obtain positional point-level information are not sensitive to the higher-order topology of the point cloud.

**Contributions** We introduce TOPF (Figure 1), a novel method to compute node-level topological features relating individual points to global topological structures of point clouds. TOPF **(i)** *outperforms* other methods and embeddings for clustering downstream tasks on topologically structured data, returns **(ii)** *provably meaningful representations*, and is **(iii)** *robust to noise*. Finally, we introduce the topological clustering benchmark suite, the first benchmark for topological clustering.

**Related Work** The intersection of topological data analysis, topological signal processing and geometry processing has many interesting related developments in the past few years. On the side of homology and TDA, the authors in [16] and [41] use harmonic *co*homology representatives to reparametrise point clouds based on circular coordinates. This implicitly assumes that the underlying structure of the point cloud is amenable to such a characterization. In [2, 26], the authors develop and use harmonic persistent homology for data analysis. However, among other differences their focus is not on providing robust topological point features. [24] uses the harmonic space of the Hodge Laplacians to cluster point clouds respecting topology, but is unstable against some form of noise, has no possibility for features selection across scales and is computationally far more expensive than TOPF. For a more in-depth review of related work, see Appendix A

**Organisation of the paper** In Section 2, we give an overview over the main ideas and concepts behind of TOPF. In Section 3, we describe how to compute TOPF. In Section 4, we give a theoretical

result guaranteeing the correctness of TOPF. Finally, we will apply TOPF on synthetic and real-world data in Section 5. Furthermore, Appendix A contains a brief history of topology and a detailed discussion of related work. Appendix B contains additional theoretical considerations, Appendix C describes the novel topological clustering benchmark suite, Appendix D contains details on the implementation and the choice of hyperparameters, Appendix E gives a detailed treatment of feature selection, Appendix F discusses simplicial weights, and Appendix G discusses limitations in detail.

## 2   Main Ideas of TOPF

A main goal of algebraic topology is to capture the shape of spaces. Techniques from topology describe globally meaningful structures that are indifferent to local perturbations and deformations. This robustness of topological features to local perturbations is particularly useful for the analysis of large-scale noisy datasets. To apply the ideas of algebraic topology in our TOPF pipeline, we need to formalise and explain the notion of *topological features*. An important observation for this is that high-dimensional point clouds and data may be seen as being sampled from topological spaces — most of the time, even low-dimensional submanifolds of $\mathbb{R}^n$ [21].

In this section we provide a broad overview over the most important concepts of topology and TDA for our context, prioritising intuition over technical formalities. The interested reader is referred to [7, 27, 49] for a complete technical account of topology and [38] for an overview over TDA.

**Simplicial Complexes**   Spaces in topology are *continuous*, consist of *infinitely* many points, and often live in *abstract space*. Our input data sets however consist of *finitely* many points embedded in *real space* $\mathbb{R}^n$. In order to bridge this gap and open up topology to computational methods, we need a notion of discretised topological spaces consisting of finitely many base points with finite description length. A *Simplicial Complex* is the simplest discrete model that can still approximate any topological space occuring in practice [43]:

**Definition 2.1** (Simplicial complexes)**.** A *simplicial complex* (SC) $\mathcal{S}$ consists of a set of vertices $V$ and a set of finite non-empty subsets (simplices, $S$) of $V$ closed under taking non-empty subsets, such that the union over all simplices $\bigcup_{\sigma \in S} \sigma$ is $V$. In the following, we will often identify $\mathcal{S}$ with its set of simplicies $S$ and denote by $\mathcal{S}_k$ the set of simplices $\sigma \in S$ with $|\sigma| = k + 1$, called *$k$-simplices*. We say that $\mathcal{S}$ is $n$-dimensional, where $n$ is the largest $k$ such that the set of $k$-simplices $\mathcal{S}_k$ is non-empty. The *$k$-skeleton* of SC contains the simplices of dimension at most $k$. If the vertices $V$ lie in real space $\mathbb{R}^n$, we call the convex hull in $\mathbb{R}^n$ of a simplex $\sigma$ its *geometric realisation* $|\sigma|$. When doing this for every simplex of $\mathcal{S}$, we call this the *geometric realisation of $\mathcal{S}$*, $|\mathcal{S}| \subset \mathbb{R}^n$.

Concretely, we can construct an $n$-dimensional SC $\mathcal{S}$ in $n + 1$ steps: First, we start with a set of vertices $V$ which we can identify with the 0-simplices $\mathcal{S}_0$. Second, we connect certain pairs of vertices with edges, which constitute the set of 1-simplices. We can then choose to fill in some triples of vertices which are fully connected by 1-simplices with triangles, i.e. 2-simplices. More generally, in the $k^{\text{th}}$ step, we can add a $k$-simplex for every set $\sigma_k$ of $k + 1$ vertices such that every $k$-element subset $\sigma_{k-1}$ of $\sigma_k$ is already a $(k - 1)$-simplex.

**Vietoris–Rips and $\alpha$-complexes**   We now need a way to construct a *simplicial complex* that approximates the *topological structure* inherent in our data set $X \subset \mathbb{R}^n$. Such a construction will always depend on the scale of the structures we are interested in. When looking from a very large distance, the point cloud will appear as a singular connected blob in the otherwise empty and infinite real space, on the other hand when we continue to zoom in, the point cloud will at some point appear as a collection of individual points separated by empty continuous space; all interesting information can be found in-between these two extreme scales where some vertices are joined by simplices and others are not. Instead of having to pick a single scale, the *Vietoris–Rips (VR) filtration* and the *$\alpha$-filtration* take as input a point cloud and return a nested sequence of simplicial complexes indexed by a scale parameter $\varepsilon$ approximating the topology of the data across all possible scales.

**Definition 2.2** (VR complex)**.** Given a finite point cloud $X$ in a metric space $(\mathcal{M}, d)$ and a non-negative real number $\varepsilon \in \mathbb{R}_{\geq 0}$, the associated VR complex $VR_\varepsilon(X)$ is given by the vertex set $X$ and the set of simplices $S = \{\sigma \subset X \mid \sigma \neq \emptyset, \forall x, y \in \sigma : d(x, y) \leq \varepsilon\}$

Intuitively, a VR complex with parameter $\varepsilon$ consists of all simplices $\sigma$ where all vertices $x \in \sigma$ have a pair-wise distance of at most $\varepsilon$. For $r \leq r'$, we obtain the canonical inclusions $i_{r,r'}(X) \colon VR_r(X) \hookrightarrow$

$VR_{r'}(X)$. The set of VR complexes on $X$ for all possible $r \in \mathbb{R}_{\geq 0}$ together with the inclusions then form the *VR filtration* on $X$. For large point clouds, using the VR complex for computations becomes expensive due to its large number of simplices. In contrast, the more sophisticated $\alpha$-complex approximates the topology of a point cloud using far fewer simplices and thus we will make use of it. For a complete account and definition of $\alpha$-complexes and our reason to use them, see Appendix B.

**Boundary matrices**  So far, we have discussed a discretised version of topological spaces in the form of SCs and a way to turn point clouds into a sequence of SCs indexed by a scale parameter. However, we still need an *algebraic representation* of simplicial complexes that is capable of encoding the structure of the SC and enables extraction of the *topological features*: The *boundary matrices* $\mathcal{B}_k$ associated to an SC $\mathcal{S}$ store all structural information of SC. The rows of $\mathcal{B}_k$ are indexed by the $k$-simplices of $\mathcal{S}$ and the columns are indexed by the $(k+1)$-simplices.

**Definition 2.3** (Boundary matrices). Let $\mathcal{S}$ be a simplicial complex and $\preceq$ a total order on its vertices $V$. Then, the $i$-th face map in dimension $n$ $f_i^n \colon \mathcal{S}_n \to \mathcal{S}_{n-1}$ is given by

$$f_i^n \colon \{v_0, v_1, \ldots, v_n\} \mapsto \{v_0, v_1, \ldots, \widehat{v_i}, \ldots, v_n\}$$

with $v_0 \preceq v_1 \preceq \cdots \preceq v_n$ and $\widehat{v_i}$ denoting the omission of $v_i$. Now, the $n$-th *boundary operator* $\mathcal{B}_n \colon \mathbb{R}[\mathcal{S}_{n+1}] \to \mathbb{R}[\mathcal{S}_n]$ with $\mathbb{R}[\mathcal{S}_n]$ being the real vector space over the basis $\mathcal{S}_n$ is given by

$$\mathcal{B}_n \colon \sigma \mapsto \sum_{i=0}^{n+1} (-1)^i f_i^{n+1}(\sigma).$$

When lexicographically ordering the simplex basis, we can view $\mathcal{B}_n$ as a *matrix*. We call $\mathbb{R}[\mathcal{S}_n]$ the space of $n$-chains. Now, $\mathcal{B}_0$ is the vertex-edge incidence matrix of the associated graph consisting of the 0- and 1-simplices of $\mathcal{S}$ and $\mathcal{B}_1$ is the edge-triangle incidence matrix of $\mathcal{S}$

**Betti Numbers and Persistent Homology**  We now turn to the notion of *topological features* and how to extract them. *Homology* is one of the main algebraic invariants to capture the shape of topological spaces and SC. From a technical point of view, the $k$-th homology module $H_k(\mathcal{S})$ of an SC $\mathcal{S}$ with boundary operators $\mathcal{B}_k$ is defined as $H_k(\mathcal{S}) \coloneqq \ker \mathcal{B}_{k-1} / \operatorname{Im} \mathcal{B}_k$. The *generator* or representative of a homology class is an element of the kernel $\ker \mathcal{B}_{k-1}$. In dimension 1, these are given by formal sums of 1-simplices forming closed loops in the SC. Importantly, the rank $\operatorname{rk} H_k(\mathcal{S})$ is called the $k$-th *Betti number* $B_k$ of $\mathcal{S}$. In dimension 0, $B_0$ counts the number of connected components, $B_1$ counts the number of loops around 'holes' of the space, $B_2$ counts the number of 3-dimensional voids with 2-dimensional boundary, and so on.

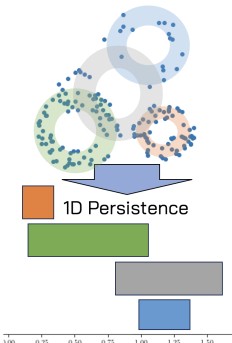

Figure 2: Sketch of Persistent Homology, [23]

If we are now given a filtration of simplicial complexes instead of a single SC, we can track how the homology modules evolve as the simplicial complex grows. The mathematical formalisation, *persistent homology*, thus turns a point cloud via a simplicial filtration into an algebraic object summarising the topological feature of the point cloud. For better computational performance, the computations are usually done in one of the small finite fields $\mathbb{Z}/p\mathbb{Z}$. Because we will later be interested in the sign of numbers to distinguish different simplex orientations, we will use $\mathbb{Z}/3\mathbb{Z}$-coefficients, with $\mathbb{Z}/3\mathbb{Z}$ being the smallest field being able to distinguish 1 and $-1$.

**The Hodge Laplacian and the Harmonic Space**  In the previous part, we have introduced a language to characterise the global shape of spaces and point clouds. However, we still need to find a way to relate these *global characterisations* back to *local properties* of the point cloud. We will do so by using ideas and concepts from differential geometry and topology: The simplicial Hodge Laplacian is a discretisation of the Hodge–Laplace operator acting on differential forms of manifolds:

**Definition 2.4** (Hodge Laplacian). Given a simplicial complex $\mathcal{S}$ with boundary operators $\mathcal{B}_k$, we define the $n$-th Hodge Laplacian $L_n \colon \mathbb{R}[\mathcal{S}_n] \to \mathbb{R}[\mathcal{S}_n]$ by setting

$$L_n \coloneqq \mathcal{B}_{n-1}^\top \mathcal{B}_{n-1} + \mathcal{B}_n \mathcal{B}_n^\top.$$

The Hodge Laplacian gives rise to the Hodge decomposition theorem:

**Algorithm 1** Topological Point Features (TOPF)

---

**Input:** Point cloud $X \in \mathbb{R}^n$, maximum homology dimension $d \in \mathbb{N}$, interpolation coeff. $\lambda$.
**1.** Compute persistent homology with generators in dimension $k \leq d$.
**2.** Select set of significant features $(b_i, d_i, g_i)$ with birth, death, and generator in $\mathbb{F}_3$ coordinates.
**3.** Embed $g_i$ into real space and project into harmonic subspace of SC at step $t = \lambda b_i + (1 - \lambda)d_i$.
**4.** Normalise projections to $\mathbf{e}_i^k$ and compute $F_k^i(x) := \text{avg}_{x \in \sigma}(\mathbf{e}_i^k l(\sigma))$ for all points $x \in X$.
**Output:** Features of $x \in X$

---

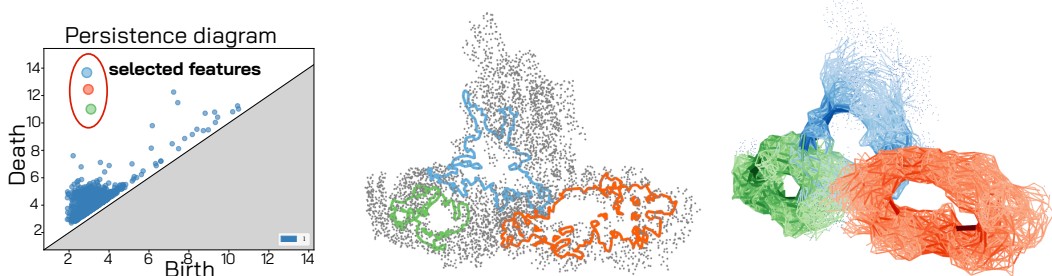

Figure 3: **TOPF pipeline applied to NALCN channelosome, a membran protein [32].** *Left:* Steps **1&2a**, when computing persistent 1-homology, three classes are more prominent than the rest. *Centre:* Step **2b**: The selected homology generators. *Right:* Step **3**: The projections of the generators into (weighted) harmonic are now each supported on one of the three rings.

**Theorem 2.5** (Hodge Decomposition [34, 46, 44])**.** *For an SC $\mathcal{S}$ with boundary matrices $(\mathcal{B}_i)$ and Hodge Laplacians $(L_i)$, we have in every dimension $k$*

$$\mathbb{R}[\mathcal{S}_k] = \underbrace{\text{Im } \mathcal{B}_{k-1}^\top}_{\text{gradient space}} \oplus \underbrace{\text{ker } L_k}_{\text{harmonic space}} \oplus \underbrace{\text{Im } \mathcal{B}_k}_{\text{curl space}} .$$

This, together with the fact that the $k$-th harmonic space is isomorphic to the $k$-th real-valued homology group $\text{ker } L_k \cong H_k(\mathbb{R})$ means that we can associate a *unique harmonic representative* to every homology class. The harmonic space encodes higher-order generalisations of smooth flow around the holes of the simplicial complex. Intuitively, this means that for every abstract global homology class of persistent homology from above we can now compute one unique harmonic representative in $\text{ker } L_k$ that assigns every simplex a value based on how much it contributes to the homology class. Thus, the Hodge Laplacian is a gateway between the *global topological features* and the *local properties* of our SC. It is easy to show that the kernel of the Hodge Laplacian is the intersection of the kernel of the boundary and the coboundary map $\text{ker } L_k = \text{ker } \mathcal{B}_{n-1} \cap \text{ker } \mathcal{B}_n^\top$. Because we have finite SCs we can identify the spaces of chains and cochains. This leads to another characterisation of the harmonic space: The space of chains that are simultaneously homology and cohomology representatives.

## 3 How to Compute Topological Point Features

In this section, we will combine the ideas and insights of the previous section to give a complete account of how to compute Topological point features (TOPF). A pseudo-code version can be found in Algorithm 1 and an overview in Figure 1. We start with a finite point cloud $X \subset \mathbb{R}^n$.

**Step 1: Computing the persistent homology** First, we need to determine the *most significant persistent homology classes* which determine the shape of the point cloud. By doing this, we can also extract the "interesting" scales of the data set. We will later use this to construct SCs to derive local variants of the global homology features. Thus we first compute the persistent $k$-homology modules $P_k$ including a set of homology representatives $R_k$ of $X$ using an $\alpha$-filtration for $n \leq 3$ and a VR filtration for $n > 3$. We use $\mathbb{Z}/3\mathbb{Z}$ coefficients to be sensitive to simplex orientations. In case we have prior knowledge on the data set, we can choose a real number $R \in \mathbb{R}_{>0}$ and only compute the filtration and persistent homology connecting points up to a distance of at most $R$. In data sets like protein atom coordinates, this might be useful as we have prior knowledge on what constitutes the

189 "interesting" scale, reducing computational complexity. See Figure 3 *left* for a persistent homology
190 diagram.

**Step 2: Selecting the relevant topological features**  We now need to select the relevant *homology*
192 *classes* which carry the most important *global information*. The persistent homology $P_k$ module in
193 dimension $k$ is given to us as a list of pairs of birth and death times $(b_i^k, d_i^k)$. We can assume these
194 pairs are ordered in non-increasing order of the durations $l_i^k = d_i^k - b_i^k$. This list is typically very
195 long and consists to a large part of noisy homological features which vanish right after they appear.
196 In contrast, we are interested in connected components, loops, cavities, etc. that *persist* over a long
197 time, indicating that they are important for the shape of the point cloud. Distinguishing between the
198 relevant and the irrelevant features is in general difficult and may depend on additional insights on
199 the domain of application.  In order to provide a heuristic which does not depend on any a-priori
200 assumptions on the number of relevant features we pick the smallest quotient $q_i^k := l_{i+1}^k / l_i^k > 0$
201 as the point of cut-off $N_k := \arg\min_i q_i^k$. The only underlying assumption of this approach is that
202 the band of "relevant" features is separated from the "noisy" homological features by a drop in
203 persistence. If this assumption is violated, the only possible way to do meaningful feature selection
204 depends on application-specific domain knowledge. We found that our proposed heuristics work well
205 across a large scale of applications. See Figure 3 *left* and *centre* for an illustration and Appendix E
206 for more technical details and ways to improve and adapt the feature selection module of TOPF. We
207 call the chosen $k$-homology classes including $k$-homology generators in dimension $f_k^i$.

**Step 3: Projecting the features into harmonic space and normalising**  In this step, we need to
209 relate the *global topology* extracted in the previous step to the simplices which we will use to compute
210 the *local* topological point feature. Every selected feature $f_k^i$ of the previous step comes with a birth
211 time $b_{i,k}$ and a death time $d_{i,k}$. This means that the homology class $f_k^i$ is present in every SC of
212 the filtration between step $\varepsilon = b_{i,k}$ and $\varepsilon = d_{i,k}$ and we could choose any of the SCs for the next
213 step. Picking a *small* $\varepsilon$ will lead to *fewer* simplices in the SC and thus to a very *localised* harmonic
214 representative. Picking a *large* $\varepsilon$ will lead to *many* simplices in the SC and thus to a very *smooth*
215 and "blurry" harmonic representative with large support. Finding a middle ground between these
216 regimes returns optimal results. For the interpolation parameter $\gamma \in (0,1)$, we will thus consider the
217 simplicial complex $\mathcal{S}^{t_{i,k}}(X)$ at step $t_{i,k} := b_{i,k}^{1-\gamma} d_{i,k}^{\gamma}$ for $k > 0$ and at step $t_{i,k} := \gamma d_{i,k}$ for $k = 0$
218 of the simplicial filtration. At this point, the homology class $f_k^i$ is still alive. We then consider the
219 real vector space $\mathbb{R}[\mathcal{S}_k^{t_{i,k}}(X)]$ with formal basis consisting of the $k$-simplices of the SC $\mathcal{S}^{t_{i,k}}$. From
220 the persistent homology computation of the first step, we also obtain a generator of the feature $f_k^i$,
221 consisting of a list $\Sigma_k^i$ of simplices $\hat{\sigma}_j \in \mathcal{S}_k^{b_{i,k}}$ and coefficients $c_j \in \mathbb{Z}/3\mathbb{Z}$. We need to turn this
222 formal sum of simplices with $\mathbb{Z}/3\mathbb{Z}$-coefficients into a vector in the real vector space $\mathbb{R}[\mathcal{S}_k^{t_{i,k}}(X)]$:
223 Let $\iota \colon \mathbb{Z}/3\mathbb{Z}$ be the map induced by the canonical inclusion of $\{-1, 0, 1\} \hookrightarrow \mathbb{R}$. We can now define
224 an indicator vector $e_k^i \in \mathbb{R}[\mathcal{S}_k^{t_{i,k}}(X)]$ associated to the feature $f_k^i$.

$$e_k^i(\sigma) := \begin{cases} \iota(c_j) & \exists \hat{\sigma}_j \in \Sigma_k^i : \sigma = \hat{\sigma}_j \\ 0 & \text{else} \end{cases}.$$

225 While this homology representative lives in a real vector space, it is not unique, has a small support,
226 and can differ largely between close simplices. All of these problems can be solved by projecting
227 the homology representative to the harmonic subspace $\ker L_k$ of $\mathbb{R}[\mathcal{S}_k^{t_{i,k}}(X)]$. Rather than directly
228 projecting $e_k^i$ to the harmonic subspace, we make use of the Hodge decomposition theorem (The-
229 orem 2.5) which allows us to compute the gradient and curl projections solving computationally
230 efficient least square problems:

$$e_{k,\text{grad}}^i := \mathcal{B}_{k-1}^\top \arg\min_{x \in \mathbb{R}[\mathcal{S}_{k-1}]} \left\| e_k^i - \mathcal{B}_{k-1}^\top x \right\|_2^2 \quad \text{and} \quad e_{k,\text{curl}}^i := \mathcal{B}_k \arg\min_{x \in \mathbb{R}[\mathcal{S}_{k+1}]} \left\| e_k^i - e_{k,\text{grad}}^i - \mathcal{B}_k x \right\|_2^2$$

231 and then setting $\hat{e}_k^i := e_k^i - e_{k,\text{grad}}^i - e_{k,\text{curl}}^i$. (Cf. Figure 3 *right* for a visualisation.) Because homology
232 representatives are gradient-free, we only need to consider the projection of $e_k^i$ into the curl space.

**Step 4: Processing and aggregation at a point level**  In the previous step, we have computed
234 a set of simplex-valued harmonic representatives of homology classes. However, these simplices
235 likely have no real-world meaning and the underlying simplicical complexes differ depending
236 on the birth and death times of the homology classes.  Hence in this step, we will collect the

features on the point-level after performing some necessary preprocessing. Given a simplex-valued vector $\hat{e}_k^i$ and a hyperparameter $\delta$, we now construct $\mathbf{e}_k^i\colon \mathcal{S}_k^{t_{i,k}}(X) \to [0,1]$ by setting $\mathbf{e}_k^i\colon \sigma \mapsto \in \{|\hat{e}_k^i(\sigma)|/(\delta \max_{\sigma' \in \mathcal{S}_k^{t_{i,k}}(X)} |\hat{e}_k^i(\sigma')|), 1\}$ such that $\hat{e}_k^i$ is normalised to $[0,1]$, the values of $[0,\delta]$ are mapped linearly to $[0,1]$ and everything above is sent to 1. We found empirically that a thresholding parameter of $\delta = 0.07$ works best across at the range of applications considered below. However, TOPF is not sensitive to small changes to $\delta$ because entries of $\hat{e}_k^i$ are concentrated around 0.

For every feature $f_k^i$ in dimension $k$ with processed simplicial feature vector $\mathbf{e}_k^i$ and simplicial complex $\mathcal{S}^{t_{i,k}}$, we define the point-level feature map $F_i^k\colon X \to \mathbb{R}$ mapping from the initial point cloud $X$ to $\mathbb{R}$ by setting

$$F_i^k\colon v \mapsto \frac{\sum_{\sigma_k \in \mathcal{S}_k^{t_{i,k}}\colon v \in \sigma_k} \mathbf{e}_k^i(\sigma_k)}{\max(1, |\{\sigma_k \in \mathcal{S}_k^t\colon v \in \sigma_k\}|)}.$$

For every point $v$, we can thus view the vector $(F_i^k(v)\colon f_i^k \in \mathcal{F})$ as a feature vector for $v$. We call this collection of features *Topological Point Features* (TOPF). (Cf. Figure 4 for an example).

**Choosing Simplicial Weights**   By default, the simplicial complexes of $\alpha$- and VR filtrations are unweighted. However, the weights determine the entries of the harmonic representatives, increasing and decreasing the influence of certain simplices and parts of the simplicial complex. We can use this observation to increase the robustness of TOPF against the influence of heterogeneous point cloud structure, which is present in virtually all real-world data sets. For a complete technical account of how and why we do this, see Appendix F.

## 4   Theoretical guarantees

In this section, we prove the relationship between TOPF and actual topological structure in datasets:

**Theorem 4.1** (Topological Point Features of Spheres). *Let $X$ consist of at least $(n + 2)$ points (denoted by $S$) sampled uniformly at random from a unit $n$-sphere in $\mathbb{R}^{n+1}$ and an arbitrary number of points with distance of at least $2$ to $S$. When we now consider the $\alpha$-filtration on this point cloud, with probability $1$ we have that **(i)** there exists an $n$-th persistent homology class generated by the $2$-simplices on the convex hull hull of $S$, **(ii)** the associated unweighted harmonic homology representative takes values in $\{0, \pm 1\}$ where the $2$-simplices on the boundary of the convex hull are assigned a value of $\pm 1$, and **(iii)** the support of the associated topological point feature (TOPF) $\mathcal{F}_n^*$ is precisely $S$: $\mathrm{supp}(\mathcal{F}_n^*) = S$. **(iv)** The same holds true for point clouds sampled from multiple $n_i$-spheres if the above conditions are met on each individual sphere.*

We will give a proof of this theorem in Appendix B.

*Remark* 4.2. In practice, datasets with topological structure consist in a majority of cases of points sampled with noise from deformed $n$-spheres. The theorem thus guarantees that TOPF will recover these structural information in an idealised setting. Experimental evidence suggests that this holds under the addition of noise as well which is plausible as harmonic persistent homology is robust against some noise [2].

## 5   Experiments

In this section, we conduct experiments on real world and synthetic data, compare the clustering results with clustering by TPCC, other classical clustering algorithms, and other point features, and demonstrate the robustness of TOPF against noise.

**Topological Point Cloud Clustering Benchmark**   We introduce the topological clustering benchmark suite (Appendix C) and report running times and the accuracies of clustering based on TOPF and other methods and point embeddings, see Table 1. We see that TOPF *outperforms* all classical clustering algorithms on all but one dataset by a wide margin. We also see that TOPF closely matches the performance of the only other higher-order topological clustering algorithm, TPCC on two datasets with clear topological features, whereas TOPF *outperforms* TPCC on datasets with more complex structure. In addition, TOPF has a consistently lower running time with better scaling for the more

Table 1: **Quantitative performance comparison of clustering with TOPF and other features/clustering algorithms.** Four $2D$ and three $3D$ data sets of the topological clustering benchmark suite (Appendix C, cf. Figure 6 for ground truth labels and Figure 7 for clustering results of TOPF). We ran each algorithm 20 times and list the mean adjusted rand index (ARI) with standard deviation $\sigma$ and mean running time. We omit $\sigma$ for algorithms with $\sigma = 0$ on every dataset. TOPF consistently outperforms or almost matches the other algorithms while having significantly better run time than the second best performing algorithm TPCC. Spectral Clustering (SC), DBSCAN, and Agglomerative Clustering (AgC) are standard clustering algorithms, ToMATo is a topological clustering algorithm [11], Geo clusters using 12-dimensional point geometric features extracted by `pgeof` and the normal point coordinates, whereas node2vec [25] produces node embeddings on a $k$-nearest neighbour graph built upon an affinity matrix. We highlight all ARI scores within $\pm 0.05$ of the best ARI score.

| | | TOPF **(ours)** | TPCC | SC | DBSCAN | AgC | ToMATo | Geo | node2vec |
|---|---|---|---|---|---|---|---|---|---|
| `4spheres` | ARI | **0.81** | 0.52±0.17 | 0.37 | 0.00 | 0.45 | 0.32 | 0.20 | 0.00±0.00 |
| | time (s) | 14.5 | 23.3 | 0.2 | 0.0 | 0.0 | 0.0 | 0.2 | 48.4 |
| `Ellipses` | ARI | **0.95** | 0.47±0.04 | 0.25 | 0.19 | 0.52 | 0.29 | 0.81 | 0.02±0.00 |
| | time (s) | 12.7 | 14.4 | 0.1 | 0.0 | 0.0 | 0.0 | 0.1 | 11.2 |
| `Spheres+Grid` | ARI | 0.70 | 0.39±0.04 | **0.90** | **0.92** | **0.89** | 0.82 | 0.41 | 0.01±0.00 |
| | time (s) | 13.0 | 28.5 | 0.5 | 0.0 | 0.0 | 0.0 | 0.3 | 63.8 |
| `Halved Circle` | ARI | **0.71** | 0.18±0.12 | 0.24 | 0.00 | 0.20 | 0.16 | 0.08 | 0.00±0.01 |
| | time (s) | 12.2 | 14.3 | 0.1 | 0.0 | 0.0 | 0.0 | 0.1 | 18.2 |
| `2Spheres2Circles` | ARI | 0.94 | **0.97**±0.01 | 0.70 | 0.00 | 0.51 | 0.87 | 0.12 | 0.00±0.00 |
| | time (s) | 38.9 | 1662.2 | 1.6 | 0.0 | 0.3 | 0.0 | 0.9 | 348.6 |
| `SphereinCircle` | ARI | 0.97 | **0.98**±0.0 | 0.34 | 0.00 | 0.29 | 0.06 | 0.69 | 0.13±0.03 |
| | time (s) | 14.5 | 8.0 | 0.0 | 0.0 | 0.0 | 0.0 | 0.08 | 20.1 |
| `Spaceship` | ARI | 0.92 | 0.56±0.03 | 0.28 | 0.26 | 0.47 | 0.30 | **0.87** | 0.07±0.00 |
| | time (s) | 16.3 | 341.8 | 16.7 | 0.0 | 0.0 | 0.0 | 0.2 | 49.8 |
| **mean** | ARI | **0.86** | 0.58 | 0.44 | 0.16 | 0.48 | 0.40 | 0.45 | 0.03 |
| | time (s) | 17.5 | 298.9 | 0.4 | 0.0 | 0.0 | 0.0 | 0.3 | 80.0 |

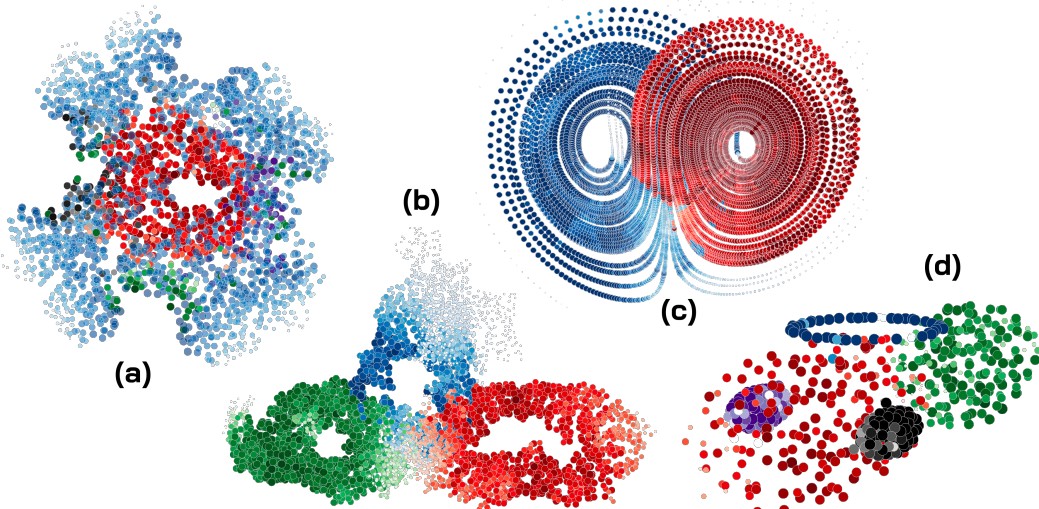

Figure 4: **TOPF on $3D$ real-world and synthetic point clouds.** For every point, we highlight the largest corresponding topological feature, where colour stands for the different features and saturation for the value of the feature. *(a):* Atoms of mutated Cys123 of E. coli [29]. We added auxiliary points on the convex hull and considered 2-homology, to detect the protein pockets which are crucial for protein-environment interactions (Cf. [40]). *(b):* Atoms of NALCN Channelosome [32] display three distinct loops. *(c):* Points sampled in the state space of a Lorentz attractor. The two features correspond to the two lobes of the attractor. *(d):* Point cloud `spaceship` of our newly introduced topological clustering benchmark suite (See Appendix C).

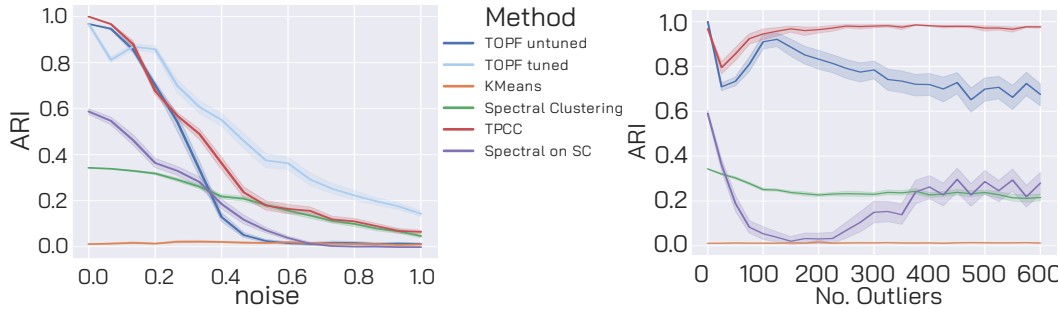

Figure 5: **Performance of Clustering based on TOPF features in increasing noise/outlier levels with 95% CI.** *Left:* We add i.i.d. Gaussian noise to every point with standard deviation indicated by the `noise` parameter. We see that even when compared with TPCC on a data set specifically crafted for TPCC, TOPF requires significantly less information and delivers almost equal performance. When tuned for datasets with a high noise level, the TOPF even outperform TPCC and drastically outperform all classical clustering algorithms. *Right:* We add outliers with the same standard deviation as the point cloud to the data set. We then measure the adjusted rand index obtained restricted on the original points. We see that even when compared with TPCC on a data set specifically crafted for TPCC, TOPF requires significantly less information and delivers matching to superior performance, significantly outperforming all other classical clustering algorithms.

complex datasets, while also not requiring prior knowledge on the best topological scale. As for the other point embeddings, Node2Vec is not able to capture any meaningful topological information, whereas the performance of clustering using geometric features depends on the data set.

**Feature Generation** In Figure 4, we show qualitatively that TOPF constructs meaningful topological features on data sets from Biology and Physics, and synthetic data, corresponding to for example rings and pockets in proteins or trajectories around different attractors in dynamical systems. (For individual heatmaps see Figure 8)

**Robustness against noise** We have evaluated the robustness of TOPF against Gaussian noise on the dataset introduced in [24] and compared the results against TPCC, Spectral Clustering, Graph Spectral Clustering on the graph constructed by TPCC, and against $k$-means in Figure 5 *Left*. We have also analysed the robustness of TOPF against the addition of outliers in Figure 5 *Right*. We see that TOPF performs well in both cases, underlining our claim of robustness.

# 6 Discussion

**Limitations** TOPF can — by design — only produce meaningful output on point clouds with a *topological structure* quantifiable by persistent homology. In practice it is thus desirable to combine TOPF with some geometric or other point-level feature extractor. As TOPF relies on the computation of persistent homology, its runtime increases on very large point clouds, especially in higher dimensions where $\alpha$-filtrations are computationally infeasible. However, subsampling, either randomly or using landmarks, usually preserves relevant topological features while improving run time [41]. Finally, selection of the relevant features is a very hard problem. While our proposed heuristics work well across a variety of domains and application scenarios, only domain- and problem-specific knowledge makes correct feature selection feasible.

**Future Work** The integration of higher-order TOPF features into ML pipelines that require point-level features potentially leads to many new interesting insights across the domains of biology, drug design, graph learning and computer vision. Furthermore, efficient computation of simplicial weights leading to the provably most faithful topological point features is an exciting open problem.

**Conclusion** We introduced point-level features TOPF founded on algebraic topology relating global structural features to local information. We gave theoretical guarantees for the correctness of their construction and evaluated them quantitatively and qualitatively on synthetic and real-world data sets. Finally, we introduced the novel topological clustering benchmark suite and showed that clustering using TOPF outperforms other available clustering methods and features extractors.

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

## A Extended Background

**A brief history of topology and machine learning** Algebraic topology is a discipline of Mathematics dating back roughly to the late 19[th] century [42]. Starting with Henri Poincaré and continuing in the early 20[th] century, the mathematical community became interested in developing a framework to capture the global shapes of manifolds and topological spaces in concise algebraic terms. This development was partly made possible by the push towards a formalisation of mathematics and analysis, in particular, which took place inside the mathematical community in the 1800's and early 1900's (e.g. [17, 30, 28]). The axiomatisation of analysis in the early 20[th] century is an important result of this process. These abstract ideas made it possible for Topologists to talk about the now common notions of Euler characteristics, Betti number, simplicial homology of manifolds, topological spaces, and simplicial and CW complexes. Over the course of the last 100 years, branching into many sub-areas like low-dimensional topology, differential topology, K-theory or homotopy theory [1, 31], algebraic topology has resolved many of the important questions and provides a comprehensive tool-box for the study of topological spaces. These achievements were tied to an abstraction and generalisation of concepts: topological spaces turned into spectra, diffeomorphism to homotopy equvialences and later weak equivalences, and Topologists turned to category theory [20], model categories [6] and recently $\infty$-categories [35] as the language of choice.

The 21[st] century saw the advent and rise of topological data analysis (TDA, [8, 12]). In short, mathematicians realised that the same notions of shape and topology that their predecessors carefully defined a century earlier were now characterising the difference between healthy and unhealthy tissue, between normal and abnormal behaviour protein behaviour, or more general between different categories in their complex data sets.

**Related Work** The intersection of topological data analysis, topological signal processing and geometry processing has many interesting related developments in the past few years. On the side of homology and TDA, the authors in [16] and [41] use harmonic *co*homology representatives to reparametrise point clouds based on circular coordinates. This implicitly assumes that the underlying structure of the point cloud is amenable to such a characterization. Although circular coordinates are orthogonal to the core goal of TOPF, the approaches share many key ideas and insights. In [2, 26], the authors develop and use harmonic persistent homology and provide a way to pool features to the point-level. However, their focus is not on providing robust topological point features and their approach includes no tunable homology feature selection across dimensions, no support for weighted simplicial complexes, and they only construct the simplicial complex at birth. In their paper on topological mode analysis, [11] use persistent homology to cluster point clouds. However, they only consider $0$-dimensional homology to base the clustering on densities and there is no clear way to generalise this to higher dimensions.

On the more geometric-centred side, [19] already provide a notion of harmonic clustering on simplices, [13, 14] analyse the notion of geometry and topology encoded in the Hodge Laplacian and its relation to homology decompositions, [45] study the normalised and weighted Hodge Laplacian in the context of random walks, and [24] use the harmonic space of the Hodge Laplacians to cluster point clouds respecting topology. Finally, a persistent variant of the Hodge Laplacian is used to study filtrations of simplicial complexes [37].

In [24], the authors have introduced TPCC, the first method to cluster a point cloud based on the higher-order topological features encoded in the data set. However, TPCC is **(i)** computationally expensive due to extensive eigenvector computations, **(ii)** depending on high-dimensional subspace clustering algorithms, which are prone to instabilities and errors, **(iii)** sensitive to the correct choice of hyperparameters, **(iv)** requiring the topological true features and noise to occur in different steps of the simplicial filtration, and it **(v)** solely focussed on clustering the points rather than extracting relevant node-level features. This paper solves all the above by completely revamping the TPCC pipeline, introducing several new ideas from applied algebraic topology and differential geometry. The core insight is: When you have the time to compute persistent homology with generators on a data set, you get the topological node features with similar computational effort.

# B  Theoretical Considerations

**More details on VR and $\alpha$-filtrations** Vietoris–Rips complexes are easy to define, approximate the topological properties of a point cloud across all scales and computationally easy to implement. However for moderately large $r$, the associated VR complex contains a large number of simplices — up to $\binom{|X|}{n}$ $n$-simplices for large enough $r$ — leading to poor computational performance for any downstream task on some large point clouds. One way to see this is the following: After adding the first edge that connects two components or the final simplex that fills a hole in the simplicial complex the VR complex keeps adding more and more simplices in the same area that keep the topology unchanged. One way to mitigate this problem is to pre-compute a set of simplices that are able to express the entire topology of the point cloud. For a point cloud $X \subset \mathbb{R}^n$, the $\alpha$-filtration consists of the intersection of the simplicial complexes of the VR filtration on $X$ with the (higher-dimensional) Delaunay triangulation of $X$ in $\mathbb{R}$. Due to algorithmic reasons, the filtration value of a simplex is then the radius of the circumscribed sphere instead of the maximum pair-wise distance of vertices. This reduces the number of required simplices across all dimensions to $O(|X|^{\lceil n/2 \rceil})$. However, the Delaunay triangulation becomes computationally infeasible for larger $n$.

**Definition B.1** ($n$-dimensional Delaunay triangulation). Given a set of vertices $V \in \mathbb{R}^n$, a Delaunay triangulation $DT(V)$ is a triangulation of $V$ such that for any $n$-simplex $\sigma_n \in DT(V)$ the interior of the circum-hypersphere of $\sigma_n$ contains no point of $DT(V)$. A triangulation of $V$ is a SC $\mathcal{S}$ with vertex set $V$ such that its geometric realisation covers the convex hull of $V$ $\text{hull}(V) = |\mathcal{S}|$ and we have for any two simplices $\sigma, \sigma'$ that the intersection of geometric realisations $|\sigma| \cap |\sigma'|$ is either empty or the geometric realisation $|\hat{\sigma}|$ of a common sub-simplex $\hat{\sigma} \subset \sigma, \sigma'$.

If $V$ is in general position, the Delaunay triangulation is unique and guaranteed to exist [18].

**Definition B.2** ($\alpha$-complex of a point cloud). Given a finite point cloud $X$ in real space $\mathbb{R}^n$, the $\alpha$-complex $\alpha_\varepsilon(X)$ is the subset of the $n$-dimensional Delaunay triangulation $DT(X)$ consisting of all $\sigma \in DT(X)$ with a radius $r$ of its circumscribed sphere with $r \leq \varepsilon$.

**Proof of the main theorem** We will now give the proof of the theorem that guarantees that TOPF works. First, let us recall Theorem 4.1:

**Theorem 4.1** (Topological Point Features of Spheres)**.** *Let $X$ consist of at least $(n + 2)$ points (denoted by $S$) sampled uniformly at random from a unit $n$-sphere in $\mathbb{R}^{n+1}$ and an arbitrary number of points with distance of at least 2 to $S$. When we now consider the $\alpha$-filtration on this point cloud, with probability 1 we have that **(i)** there exists an $n$-th persistent homology class generated by the 2-simplices on the convex hull hull of $S$, **(ii)** the associated unweighted harmonic homology representative takes values in $\{0, \pm 1\}$ where the 2-simplices on the boundary of the convex hull are assigned a value of $\pm 1$, and **(iii)** the support of the associated topological point feature (TOPF) $\mathcal{F}_n^*$ is precisely $S$: $\mathrm{supp}(\mathcal{F}_n^*) = S$. **(iv)** The same holds true for point clouds sampled from multiple $n_i$-spheres if the above conditions are met on each individual sphere.*

*Proof.* Assume that we are in the scenario of the theorem. Now because the $n$-volume of $(n-1)$-submanifolds is zero, we have that with probability 1 the points of $S$ don't lie on a single $(n-1)$ sphere inside the $n$-sphere. Let us now look at the $\alpha$-filtration of the simplices in $S$: Recall that the filtration values of a $k$-simplex is given by the radius of the $(k-1)$-sphere determined by its vertices. Because all of the $(n+1)$-simplices $\sigma_{n+1}$ with vertices $V \subset S$ in $S$ lie on the same unit $n$-sphere $S_n$, they all share the filtration value of $\alpha(\sigma_{n+1}) = 1$. By the same argument as above, with probability 1 there are no $(n+1)$ points in $S$ that lie on an *unit* $(n-1)$-sphere. Thus all of the $n$-simplices $\sigma_n$ lie on $(n-1)$-spheres $S_n$ with a radius $r < 1$ smaller than 1 and hence have a filtration value $\alpha(\sigma_n)$ smaller than 1. Let

$$b := \max\left(\{\alpha(\sigma_n) : \sigma_n \subset \partial \operatorname{hull}(S)\}\right)$$

be the maximum filtration value of an $n$-simplex on the boundary of the convex hull of $S$. Then, then a linear combination $g$ of the $n$-simplices of the boundary of the convex hull of $S$ with coefficients in $\pm 1$ is a generator of a persistent homology class with life time $(b, 1)$ (this follows from the fact that $n$-spheres and their triangulations are orientable). This proves claim **(i)**.

Because of the assumption that all points not contained in $S$ have a distance of at least 2 to the points in $S$, all $(n+1)$-simplices $\sigma_{n+1}$ with vertices both in $S$ and its complement in $X$ will have a filtration value $\alpha(\sigma_{n+1}) \geq 1$ of at least 1. Recall that all $(n+1)$-simplices $\sigma_{n+1} \subset S$ with vertices inside $S$ have a filtration value of $\alpha(\sigma_{n+1}) = 1$. Thus the adjoint of the $n$-th boundary operator $\mathcal{B}_n^\top$ is trivial on the homology generator $g$. Thus, we have that for the $n$-th Hodge Laplacian

$$L_n g = \mathcal{B}_{n-1}^\top \mathcal{B}_{n-1} g + \mathcal{B}_n \mathcal{B}_n^\top g = 0 + 0 = 0$$

and hence $g$ is a harmonic generator for the entire filtration range of $(b, 1)$, which proves claim **(ii)**. Claim **(iii)** and **(iv)** then follow from the construction of the TOPF values. $\square$

## C  Topological Clustering Benchmark Suite

We introduce seven point clouds for topological point cloud clustering in the topological clustering benchmark suite (TCBS). The ground truth and the point clouds are depicted in Figure 6. The point clouds represent a mix between 0-, 1- and 2-dimensional topological structures in noiseless and noisy settings in ambient 2-dimensional and 3-dimensional space. The results of clustering according to TOPF can be found in Figure 7.

## D  Implementation

We will release an implementation of TOPF and the code and data required to reproduce the experimental results of this paper under `https://anonymous.4open.science/r/topf_submission-5C40/`. In particular, we will release the topological clustering benchmark suite.

All experiments were run on a Apple M1 Pro chipset with 10 cores and 32 GB memory. TOPF and the experiments are implemented in Python and Julia. For persistent homology computations, we used GUDHI [48] (© The GUDHI developers, MIT license) and Ripserer [52] (© mtsch, MIT license), which is a modified Julia implementation of [3]. For the least square problems, we used the LSMR implementation of SciPy [22]. We used the Node2Vec python implementation `https://github.com/eliorc/node2vec` (© Elior Cohen, MIT License) based on the Node2Vec Paper [25]. We used the `pgeof` Python package for computation of geometric features `https://github.`

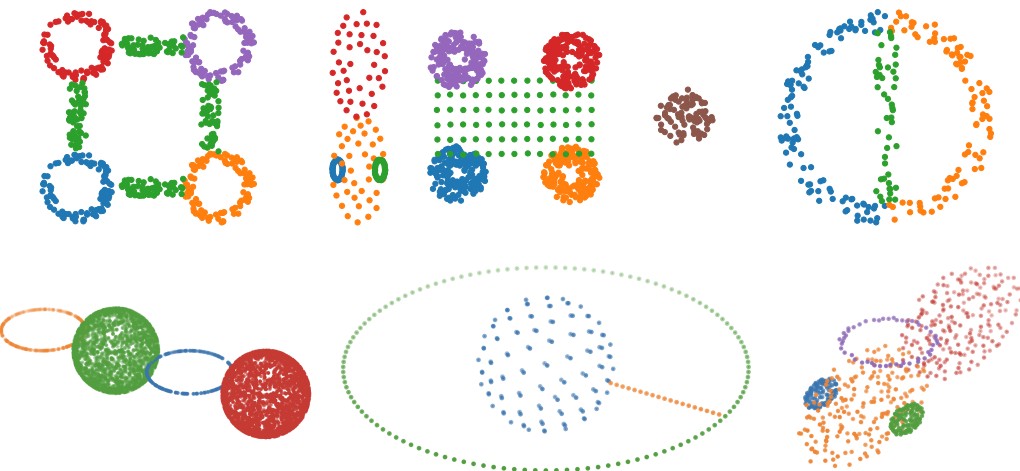

Figure 6: **Data sets of the Topological Clustering Benchmark Suite (TCBS) with true labels.** *Top:* $2D$ data sets. *From left to right:* 4Spheres (656 points), Ellipses (158 points), Spheres+Grid (866 points), Halved Circle (249 points). *Bottom:* $3D$ data sets. *From left to right:* 2Spheres2Circles (4600 points), SphereinCircle (267 points), spaceship (650 points).

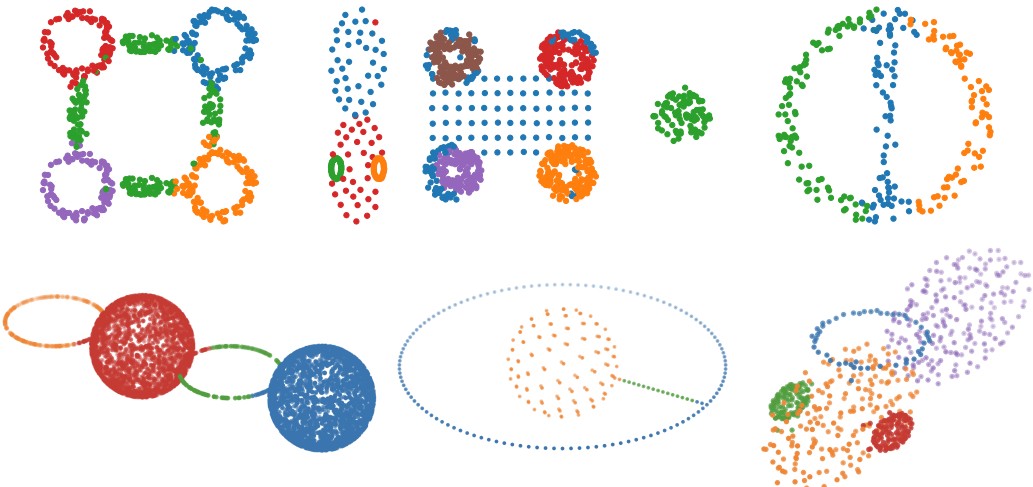

Figure 7: **Data sets of the Topological Clustering Benchmark Suite (TCBS) with labels generated by TOPF.** *Top:* $2D$ data sets. *From left to right:* 4Spheres (0.81 ARI), Ellipses (0.95 ARI), Spheres+Grid (0.70 ARI), Halved Circle (0.71 ARI). *Bottom:* $3D$ data sets. *From left to right:* 2Spheres2Circles (0.94 ARI), SphereinCircle (0.97 ARI), spaceship (0.92 ARI).

com/drprojects/point_geometric_features (© Damien Robert, Loic Landrieu, Romain Janvier, MIT license). We use parts of the implementation of TPCC https://git.rwth-aachen.de/netsci/publication-2023-topological-point-cloud-clustering (© Computational Network Science Group, RWTH Aachen University, MIT license).

## D.1 Hyperparameters

All the relevant hyperparameters are already mentioned in their respective sections. However, for convenience we gather and briefly discuss them in this section. We note that TOPF is robust and applicable in most scenarios when using the default parameters without tuning hyperparameters. The hyperparameters should more be thought of as an additional way where detailed domain-knowledge can enter the TOPF pipeline.

**Maximum Homology Dimension** $d$    The maximum homology dimension determines the dimensions of persistent homology the algorithm computes.

For the choice of the maximum homology degree $d$ to be considered there are mainly three heuristics which we will list in decreasing importance (Cf. [24]):

    I. In applications, we usually know which kind of topological features we are interested in, which will then determine $d$. This means that 1-dimensional homology and $d = 1$ suffices when we are looking at loops of protein chains. On the other hand, if we are working with voids and cavities in 3d histological data, we need $d = 2$ and thus compute 2-dimensional homology.

    II. Algebraic topology tells us that there are no closed $n$-dimensional submanifolds of $\mathbb{R}^n$. Hence their top-homology will always vanish and all interesting homological activity will appear for $d < n$.

    III. In the vast majority of cases, the choice will be between $d = 1$ or $d = 2$ because empirically there are virtually no higher-dimensional topological features in practice.

In our quantitative experiments, we have always chosen $d = n - 1$.

**Thresholding parameter** $\delta$    In step 4 of the algorithm, we normalise and threshold the harmonic representatives. After normalising, the entries of the vectors lie in the interval of $[0, 1]$. The thresholding parameter $\delta$ now essentially determines an interval of $[0, \delta]$ which we will linearly map to $[0, 1]$, while mapping all entries above $\delta$ to 1 as well. This is necessary as most of the entries in the vector $e_k^i$ are very close to 0 with a very small number of entries being close to 1. Without this thresholding, TOPF would now be almost entirely determined by these few large values. Thus this step limits the maximum possible influence of a single entry. However, because most of the entries of $e_k^i$ are concentrated around 0, small changes in $\delta$ will not have a large effect and we chose $\delta = 0.07$ in all our experiments.

**Interpolation coefficient** $\lambda$    The interpolation coefficient $\lambda \in [0, 1)$ determines whether we build our simplicial complexes close to the birth or the death of the relevant homological features at time $t = b^{1-\lambda}d$. This then in turns controls how localised or smooth the harmonic representative will be. In general, the noisier the ground data is the higher we should choose $\lambda$. However, TOPFis not sensitive to small changes in $\lambda$. We have picked $\lambda = 0.3$ for all the quantitative experiments, which empirically represents a good choice for a broad range of applications.

**Feature selection factor** $\beta$    Increasing $\beta$ leads to TOPF preferring to pick a larger number of relevant topological features. Without specific domain-knowledge, $\beta = 0$ represents a good choice.

**Feature selection quotients** `max_total_quot,` `min_rel_quot,` **and** `min_0_ratio`    These are technical hyperparameters controlling the feature selection module of TOPF. For a technical account of them, see Appendix E. In most of the cases without domain knowledge, they do not have an effect on the performance of TOPF and should be kept at their default values.

**Simplicial Complex Weights**    Although the simplicial weights are not technically a hyperparameter, there are many potential ways to weigh the considers SCs that can highlight or suppress different topological and geometric properties. In all our experiments, we use $w_\Delta$ weights discussed in Appendix F.

# E    How to pick the most relevant topological features

**Simplified heuristic**    The persistent homology $P_k$ module in dimension $k$ is given to us as a list of pairs of birth and death times $(b_i^k, d_i^k)$. We can assume these pairs are ordered in non-increasing order of the durations $l_i^k = d_i^k - b_i^k$. This list is typically very long and consists to a large part of noisy homological features which vanish right after they appear. In contrast, we are interested in connected components, loops, cavities, etc. that *persist* over a long time, indicating that they are important for the shape of the point cloud. Distinguishing between the relevant and the irrelevant features is in general difficult and may depend on additional insights on the domain of application. In order to provide a heuristic which does not depend on any a-priori assumptions on the number of relevant

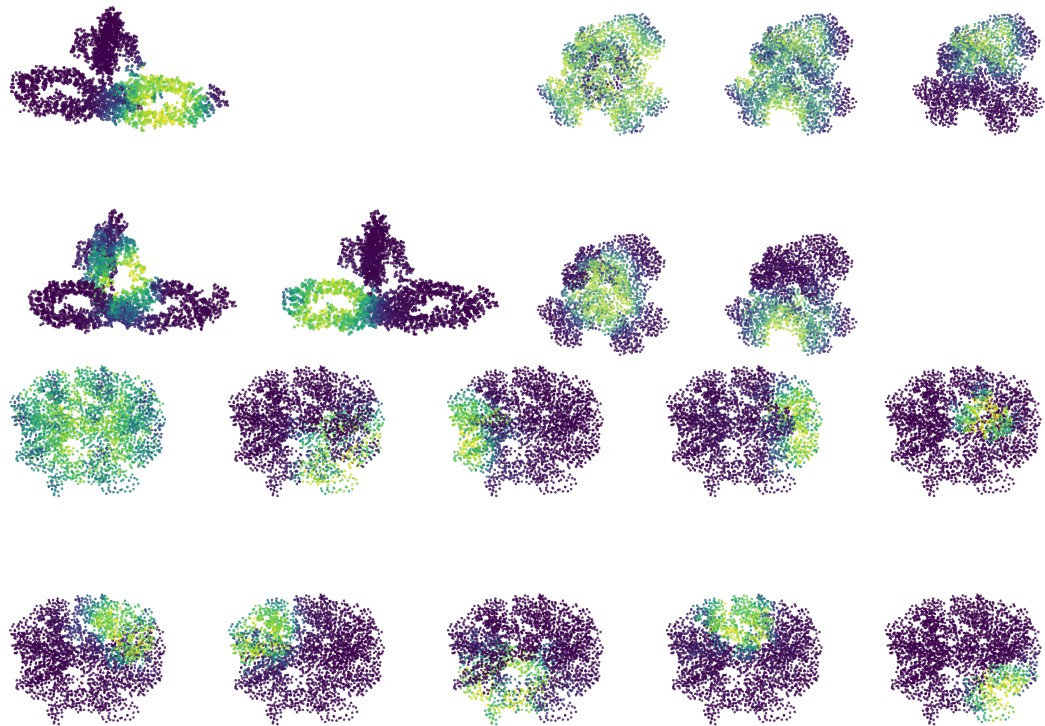

Figure 8: **TOPF heatmaps for three proteins.** *Top left* NALCN channelosome [32] *Top right:* Mutated Cys123 of E. coli [29], with convex hull added during computation, only 2-dimensional homology features *Bottom:* GroEL of E. coli [10] (Selected features).

features we pick the smallest quotient $q_i^k := l_{i+1}^k/l_i^k > 0$ as the point of cut-off $N_k := \arg\min_i q_i^k$. The only underlying assumption of this approach is that the band of "relevant" features is separated from the "noisy" homological features by a drop in persistence.

**Advanced Heuristic** However, certain applications have a single very prominent feature, followed by a range of still relevant features with significantly smaller life times, that are then followed by the noisy features after another drop-off. This then could potentially lead the heuristic to find the wrong drop-off. We propose to mitigate this issue by introducing a hyperparameter $\beta \in \mathbb{R}_{>0}$. We then define the $i$-th importance-drop-off quotient $q_i^k$ by

$$q_i^k := l_{i+1}^k/l_i^k \left(1 + \beta/i\right).$$

The basic idea is now to consider the most significant $N_k$ homology classes in dimension $k$ when setting $N_k$ to be

$$N_k := \arg\min_i q_i^k.$$

Increasing $\beta$ leads the heuristic to prefer selections with more features than with fewer features. Empirically, we still found $\beta = 0$ to work well in a broad range of application scenarios and used it throughout all experiments. There are only a few cases where domain-specific knowledge could suggest picking a larger $\beta$.

To catch edge cases with multiple steep drops or a continuous transition between real features and noise, we introduce two more checks: We allow a minimal $q_i^k$ of `min_rel_quot` $= 0.1$ and a maximal quotient $q_1^h/q_i^k$ of `max_total_quot` $= 10$ between any homology dimensions. Because features in $0$-dimensional homology are often more noisy than features in higher dimensions, we add a minimum zero-dimensional homology ratio of `min_0_ratio` $= 5$, i.e. every chosen $0$-dimensional feature needs to be at least `min_0_ratio` more persistent then the minimum persistence of the higher-dimensional features. Because these hyperparameters only deal with the edge cases of feature selection, TOPF is not very sensitive to them. For all our experiments, we used the above hyperparameters. We advise to change them only in cases where one has in-depth domain knowledge about the nature of relevant topological features.

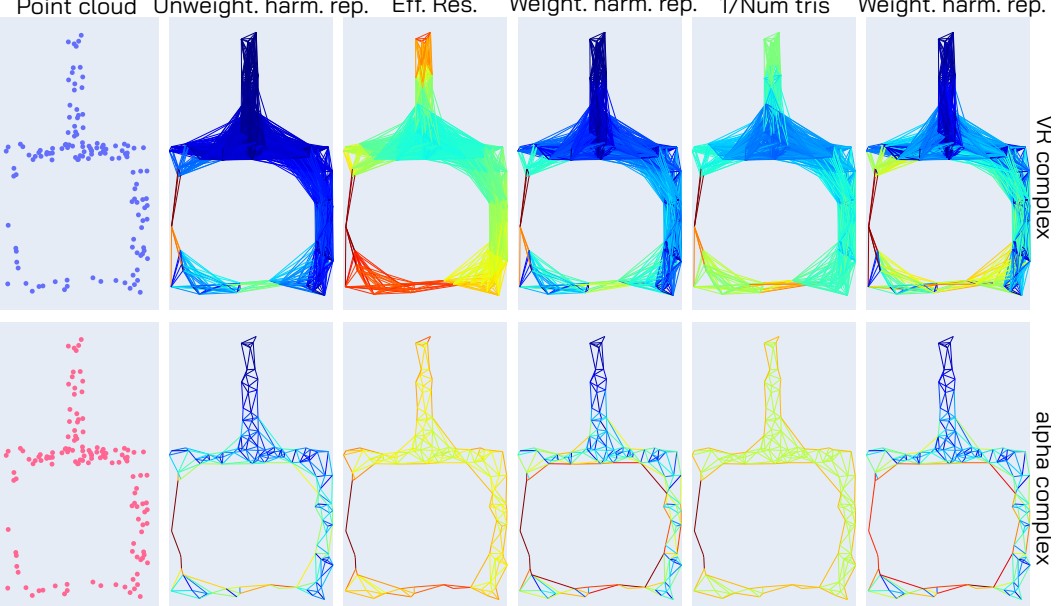

Figure 9: **Effect of weighing a simplicial complex on harmonic representatives.** *Top:* VR complex. *Bottom:* $\alpha$-complex *Left:* The base point cloud with different densities. *2nd Left:* Unweighted harmonic homology representative of the large loop. *3rd Right:* Effective resistance of the 1-simplices. *3rd Right:* Harmonic homology representative of the complex weighted by effective resistance. *2nd Right:* Inverse of number of incident triangles (Definition F.1). *Right:* Harmonic homology representative of the complex weighted by number of incident triangles. Up to a small threshold, the standard harmonic representative in the VR complex is almost exclusively supported in the low-density regions of the simplicial complex. This leads to poor and unpredictable classification performance in downstream tasks. In contrast, the harmonic homology representative of the weighted VR complex has a more homogenous support along the loop, while still being able to discriminate the edges not contributing to the loop. The $\alpha$-complex suffers less from this phenomenon (at least in dimension 2), and hence reweighing is not necessarily required.

## F    Simplicial Weights

In an ideal world, the harmonic eigenvectors in dimension $k$ would be vectors assigning $\pm 1$ to all $k$-simplices contributing to $k$-dimensional homological feature, a $0$ to all $k$-simplices not contributing or orthogonal to the feature, and a value in $(-1, 1)$ for all simplices based on the alignment of the simplex with the boundary of the void. However, this is not the case: In dimension 1, we can for example imagine a total flow of 1 circling around the hole. This flow is then split up between all parallel edges which means *two* things: **I** Edges where the loop has a *larger diameter* have *smaller harmonic values* than edges in thin areas and **II** in VR complexes, which are the most frequently used simplicial complexes in TDA, edges in areas with a *high point density* have *smaller harmonic values* than edges in low-density areas. Point **II** is another advantage of $\alpha$-complexes: The expected number of simplices per point does not scale with the point density in the same way as it does in the VR complex, because only the simplices of the Delaunay triangulation can appear in the complex.

We address this problem by weighing the $k$-simplices of the simplicial complex. The idea behind this is to weigh the simplicial complex in such a way that it increases and decreases the harmonic values of some simplices in an effort to make the harmonic eigenvectors more homogeneous. For weights $w \in \mathbb{R}^{S_k}$, $W = \text{diag}(w)$, the symmetric weighted Hodge Laplacian [45] takes the form of

$$L_k^w = W^{1/2} \mathcal{B}_{k-1} \mathcal{B}_{k-1}^\top W^{1/2} + W^{-1/2} \mathcal{B}_k \mathcal{B}_k^\top W^{-1/2}.$$

Because we want the homology representative to lie in the weighted gradient space, we have to scale its entries with the weight and set $e^i_{k,w} := W^{-1/2} e^i_k$. With this, we have that

$$\mathcal{B}^\top_{k-1} W^{1/2} e^i_{k,w} = \mathcal{B}^\top_{k-1} W^{1/2} W^{-1/2} e^i_k = \mathcal{B}^\top_{k-1} e^i_k = 0$$

We propose two options to weigh the simplicial complex. The first option is to weigh a $k$-simplex by the square of the number of $k + 1$-simplices the simplex is contained in:

$$w_\Delta(\sigma_k) = 1/(|\{\sigma_{k+1} \in \mathcal{S}^t_{k+1} : \sigma_k \subset \sigma_{k+1}\}| + 1)^2$$

where the $+1$ is to enforce good behaviour at simplices that are not contained in any higher-order simplices. One of the advantages of the $\alpha$-complex is that we don't have large concentrations of simplices in well-connected areas. The proposed weighting $w_\Delta$ is computationally straightforward, as it can be obtained as the column sums of the absolute value of the boundary matrix $|\mathcal{B}_k|$. The weights also deal with the previously mentioned problem **II**: As the homology representative is scaled inversely to the weight vector $w$, the simplices in high-density regions will be assigned a low weight and thus their weighted homology representative will have a larger entry. By the projection to the orthogonal complement of the curl space, this large entry is then diffused among the high-density region of the SC with many simplices, whereas the lower entries of the simplices in low-density regions are only diffused among fewer adjacent simplices.

However, the first weight is not able to incorporate the number of parallel simplices into the weighting. This is why we propose a second simplicial weight function based on generalised effective resistance.

**Definition F.1** (Effective Hodge resistance weights)**.** For a simplicial complex $\mathcal{S}$ with boundary matrices $(\mathcal{B}_k)$, we define the effective Hodge resistance weights $w_R$ on $k$-simplices to be:

$$w_R := \mathrm{diag}\left(\mathcal{B}^+_{k-1} \mathcal{B}_{k-1}\right)^2$$

where $\mathrm{diag}(-)$ denotes the vector of diagonal entries and $(-)^+$ denotes taking the Moore–Penrose inverse.

Intuitively for $k = 1$, we can assume that every edge has a resistance of 1 and then the effective resistance coincides with the notion from Physics. Thus simplices with many parallel simplices are assigned a small effective resistance, whereas simplices with few parallel simplices are assigned an effective resistance close to 1. However, computing the Moore–Penrose inverse is computationally expensive and only feasible for small simplicial complexes.

In Figure 9, we show that the weights $w_\Delta$ are a good approximation of the effective resistance in terms of the resulting harmonic representative. The standard form of TOPF used in all experiments uses $w_\Delta$-weights.

## G   Limitations

**Topological features are not everywhere**   The proposed topological point features take relevant persistent homology generators and turn these into point-level features. As such, applying TOPF only produces meaningful results on point clouds that have a topological structure. On these point clouds, TOPF can extract structural information unobtainable by non-topological methods. Although TDA has been successful in a wide range of applications, a large number of data sets does not possess a meaningful topological structure. Applying TOPF in these cases will produce no additional information. Other data sets require pre-processing before containing topological features. In Figure 4 *left*, the $2d$ topological features characterising protein pockets of Cys123 only appear after artificially adding points sampled on the convex hull of the point cloud (Cf [40]).

**Computing persistent homology can be computationally expensive**   As TOPF relies on the computation of persistent homology including homology generators, its runtime increases on very large point clouds. This is especially true when using VR instead of $\alpha$-filtrations, which become computationally infeasible for higher-dimensional point clouds. Persistent homology computations for dimensions above 2 are only feasible for very small point clouds. Because virtually all discovered relevant homological features in applications appear in dimension 0, 1, or 2, this does not present a large problem. Despite these computational challenges, subsampling, either randomly or using landmarks, usually preserves relevant topological features and thus extends the applicability of TDA in general and TOPF even to very large point clouds.

**Automatic feature selection is difficult without domain knowledge** While the proposed heuristics works well across a variety of domains and application scenarios, only domain- and problem-specific knowledge makes truthful feature selection feasible.

**Experimental Evaluation** There are no benchmark sets for topological point features in the literature, which makes benchmarking TOPF not straightforward. On the level of clustering, we introduced the topological clustering benchmark suite to make quantitative comparisons of TOPF possible, and benchmarked TOPF on some of the point clouds of [24]. On both the level of point features and real-world data sets, it is however hard to establish what a *ground truth* of topological features would mean. Instead we chose to qualitatively report the results of TOPF on proteins and real-world data, see Figure 4.

