# OpenReview forum: "Node-Level Topological Representation Learning on Point Clouds"
_NeurIPS.cc/2024/Conference — Submitted to NeurIPS 2024_

### Official Review · Reviewer_Zmis · 2024-07-11

**Soundness:** 3
**Presentation:** 3
**Contribution:** 2
**Rating:** 5
**Confidence:** 4

**Summary:**

Large scale topological descriptors of data are leveraged to compute point/node-level descriptors, which encode to which large scale topological feature each point belongs to. For this, a combination of applied algebraic topology and applied harmonic analysis is used. More specifically, large scale homological features are computed using persistent homology, then represented with harmonic cocyles, and then averaged locally to obtain a point-level descriptors. The problem of topological clustering (already introduced in the literature) is addressed, whose objective is to determine to which large scale topological feature a certain data point belongs to. A set of benchmark datasets are introduced for topological clustering. The pipeline is applied to these datasets as well as real world datasets.

**Strengths:**

- Large scale topology of data is leveraged to assign point/node-level features to data. This gives concrete meaning to what it means for a data point to belong to a certain large scale topological feature.
- The method is based on well-established mathematical concepts.
- The concept of topological clustering is interesting and has potential.
- A suite of synthetic datasets is introduced.

**Weaknesses:**

Regarding unjustified claims:

- Existing approaches are undersold. Specifically, in the introduction it is said that "none of these approaches is able to represent higher-order topological information" and that "such higher-order topological information is however invisible to standard tools of data analysis like PCA of k-means clustering". However, cluster structure is topological structure. Does "higher order" mean homology in dimensions 1 and above?
- Remark 4.2 says that "datasets with topological structure consist in a majority of cases of points sampled with noise from deformed n-spheres". This seems like a really strong claim. Is there any evidence of this?

Regarding theory:

- Theorem 4.1 applies in a very restricted scenario. Moreover, I do not understand why the harmonic representative takes values in {0, -1, 1}. This seems very surprising since harmonic cycles/cocycles almost always take fractional values (in order to minimize energy). I did not understand the proof of this fact; specifically, why $g$ being a harmonic generator for the entire filtration range of $(b,1)$ implies this claim.

Regarding the methodology:

- The method, specifically line 225, seems to assume that a cycle with coefficients in $Z/3Z$ will also be a cycle when interpreting those coefficients (0,-1, or 1) as real numbers. However, this need not be the case. To see this it suffices to consider a simplicial complex given by a triangle with no interior. Thus, step 3 of Algorithm 1 (and the method more generally) seems to be heuristic.
- The setup up Table 1 is unclear to me. How can one compare TOPF, which produces feature vectors, with, say, DBSCAN, which produces a clustering?
- Figure 4 is hard to interpret. For example, how should one assess the effectiveness of the algorithm in Fig 4(a)?
- The methodology has many hyperparameters. Some choices, like delta=0.07 in line 241, seem arbitrary.

**Questions:**

1. Does your method have an interpretation in the case of a Riemannian manifold? That is, suppose that I take a Riemannian manifold and a harmonic (smooth) cocycle. Is there a corresponding point-level feature function? Does it have an interpretation? Perhaps it is related to the pointwise norm of the harmonic cocycle?
2. How were the parameters of the other algorithms in table 1 selected?
3. Which further applications do you have in mind? Figure 4 is interesting, but does not hint at real life applications.

**Limitations:**

- The experimental evaluation is limited.

---

> ### Author Rebuttal · Authors · 2024-08-07
>
> We sincerely thank the reviewer for their thorough review and feedback! We believe we have significantly improved the paper based upon your comments! We will address the individual points:
>
> # "Regarding unjustified claims":
> 1. By higher-order we mean of order 1 and above in the sense of homology. In this interpretation, our claims are true and we are not underselling existing approaches, as cluster structure is only 0-th order homology structure. However, we want to be as clear and fair as possible. We have revised the sentences for clarity and state the differences more clearly.
>
> 2. This is not quite a strong claim as it might seem. Ordinary cluster structures are basically captured by noisy 0-spheres, 1-dimensional homology appears almost always in the form of some noisy deformed cycle, and so on. Even very basic non-sphere structures like tori appear only in very rare circumstances in real-world data, as is evidenced by the existing applied TDA literature. Finally, we explicitly did not claim this for all datasets and thus feel justified.
>
> ### Regarding Theory
> 1. It is true that Theorem 4.1 applies in a restricted scenario. The purpose of Thm 4.1 is to verify that TOPF does __provably correct things__ on ideal data sets recovering the implanted topological structure.
> 2. Thank you very much for reading the proof! :)
> Harmonic cycles (as those considered by topf) indeed take absolute values between 0 and 1 almost always on the majority of the simplices. This is, unless they __don't have parallel simplices around the same generalised hole__: In this case, it is not possible to minimise the energy by distributing the "flow" (in the 1-dimensional case) among the parallel simplices and a single simplex needs to account for all the contribution to homology. This is exactly the case in the theorem: Because the points lie on a __perfect n-sphere with radius 1, the first n-simplex appears only at point 1 in the alpha-filtration__. (In a VR filtration, this would not hold.) Without any n-simplices, there are __no "parallel" (n-1)-simplices__, and thus every simplex has either __value -1 or 1__ (depending on orientation) or 0 (Or a normalised version of this). We will carefully describe this argument in the proof.
>
> ### Regarding the methodology
> * The example given by the reviewer does not underline their claim: In Z/3Z coefficients, a homology representative for the triangle needs to assign every edge a value such that the oriented sum (*+1 at the tip of the edge and *-1 at the tail) at each of the nodes is equal to 0. If we fix the orientations of the edges to all be clockwise, this is only possible with all edges having the same value. All edges having a 0 obviously does not work, so (1,1,1) and (-1,-1,-1) remain as options. In the same setting, these are generators for real-valued homology as well.
>
> In general, it is true that being a generator of Z/p-homology does not guarantee being a generator of homology with real coefficients.
> However, there are multiple reasons why this is not a problem in practice.
> 1. The homology generators computed by persistent homology have a very special structure. In dimension 1, instead of being some random formal sum of edges, they form a sum of closed cycles. In higher dimensions, similar statements hold. This however then means that the PH generators in Z/3Z coefficients will always be in the kernel of the boundary operator in R coefficients.
> 2. Now the only thing that can happen is that the the generator is in the image of the boundary in R coefficients. This is precisely the case when the represented homology class has p-torsion. On real-world data, this very rare. (This is mainly due to the fact that manifold needs to be "very complicated" to have torsion, whereas torsion homology groups are easy to achieve. E.g.,  RP^2 (the most simple space with torsion in the homology group) or can only be embedded in R^4 or above).
> 3. Furthermore, it does not suffice for the space to have a homology class with some torsion, only homology classes with 3-torsion will be relevant.
> 4. We can very easily check whether this is the case: If the projection of the generator to the harmonic space, which we compute anyway, vanishes, the considered homology class has torsion, and thus we can safely exclude it from our analyses. We will add a check for this in our implementation.
> 5. We are not throwing any harmonic information away this way: all homology classes representable by harmonic forms don't have torsion (in Z-coefficients), and all homology classes without torsion appear in Z/3Z homology.
>
> We will add a version of this explanation to the appendix, thank you very much for pointing this out!
>
> * We run spectral clustering on the feature vectors produced by TOPF to obtain a clustering of the points. (k-means or similar clustering algorithms would work as well.) We have added an explanation to the experiments section and the caption of Table 1. Thank you for catching this!
>
> * The purpose of Fig. 4 is to show that TOPF corresponds roughly to our intuition of relevant topology: In a), the features detect some of the pockets (green, black, purple) of the protein and the hole in the middle (red), in b) the loops in the protein structure are recovered, and in c) the two regimes of the Lorentz attractor are recovered. Because there is no ground truth for this, we can't report an ARI, which we did in the quantitative experiments.
>
> * We have added experiments on the performance of TOPF on TCBS for a wide range of hyperparameter settings. (Fig. 1 of the pdf).
> In summary, TOPF is not very sensitive to hyperparameter changes around the default values, making hyperparameter choices robust. We have picked the hyperparameters for TOPF to perform well across the large range of applications in our paper. We will add the experiments to the appendix section discussing the hyperparameters.
>
> We apologise, but we will answer the remaining questions in a comment.

---

> ### Comment · Reviewer_Zmis · 2024-08-08
>
> Thank you for your responses.
>
> - "Harmonic cycles (as those considered by topf) indeed take absolute values between 0 and 1 almost always on the majority of the simplices." This is not what your theorem says. It says that it takes values in {$0,\pm 1$}, which still seems wrong to me.
> - "parallel simplices around the same generalised hole". What is a "generalized whole"? I did not understand this part of the discussion.
> - "We run spectral clustering on the feature vectors produced by TOPF". How were the parameters selected?
> - I am not convinced by your argument for lifting Z/3Z cocycles to Z-cocycles. Indeed, the original paper that proposed to get circular coordinates out of persistent homology computations [1] has a whole section on this, "2.4 Lifting to Integer Coefficients". In particular, they say that when the "easy" lift that you use fails, they don't have particular guidance as to how to proceed: "This is all very well. Unfortunately, the equation η = d1ζ is a Diophantine linear system. At present, we can provide no particular guidance as to how to solve the system (other than by vague appeal to off-the-shelf Diophantine or integer linear programming solvers), even if we know that a solution exists." Since the publication of that paper, the software DREiMac has been published. They do address this issue using a linear programming solver. See the documentation at [2], where they also have an example showing how things can fail. In particular, that example shows that torsion is not the only issue that can occur, as that data has no torsion.
>
>
> [1] Persistent Cohomology and Circular Coordinates. Vin de Silva, Dmitriy Morozov & Mikael Vejdemo-Johansson. https://link.springer.com/article/10.1007/s00454-011-9344-x
>
> [2] https://dreimac.scikit-tda.org/en/latest/notebooks/parameters_prime_and_check_cocycle_condition.html

---

> > ### Author Response · Authors · 2024-08-11
> >
> > Thank you very much for your in-depth response! This is incredibly helpful! We will reply to your comments below.
> >
> > > "Harmonic cycles (as those considered by topf) indeed take absolute values between 0 and 1 almost always on the majority of the simplices." This is not what your theorem says. It says that it takes values in {$0,\pm 1$}, which still seems wrong to me.
> >
> > We should have been more precise in our previous reply. Consider a k-homology representative r with simplex-values in {-1,0,1}. We consider the harmonic projection h of r into the harmonic space. Then, for a k-simplex $\sigma$, we have that $h(\sigma)=\pm 1$ iff $r(\sigma) = 1$ AND $\sigma$ is not the face of any $k+1$-simplices.
> >
> > This can for example be seen by representing h as the difference of $r$ and its gradient and curl parts $h=r-r_{grad}-r_{curl}$. Because r is already a cycle, $B_kr=0$ and thus $r_{grad}=0$. Now, the curl part can be written as stemming from a signal on the the $k+1$-simplices, $r_curl=B_{k+1}x_{curl}$. However, because $\sigma$ is not the face of a $k+1$-simplex, $r_{curl}(\sigma)=0$ and thus $h(\sigma) = r(\sigma)$
> >
> > The setting of the theorem precisely constructs a case where we have $k$-simplices, but no $(k+1)$. This is however an idealised setting. Because we construct the simplicial complex to compute the harmonic representative somewhere in the middle (determined by the interpolation coefficient) between the birth and death time of the homology class, empirically the majority of the $k$- simplices of the $k$-homology generators are faces of $(k+1)$-simplices. In this case, the harmonic representative smoothes out the feature.
> >
> > > "parallel simplices around the same generalised hole". What is a "generalized whole"? I did not understand this part of the discussion.
> >
> > We apologise for using imprecise terminology in the reply. With parallel simplices we mean $k$-simplices that are connected via a number of $(k+1)$-simplices. "Around the same generalised hole" is just a picture for them being part of the same harmonic representative of a homology class.
> >
> > > "We run spectral clustering on the feature vectors produced by TOPF". How were the parameters selected?
> >
> > We use the default parameters suggested by the scikit-learn. In case the number of clusters is known, we pass this to TOPF. If not, the number of selected topological features in the previous step is passed as n_clusters.
> >
> > > I am not convinced by your argument for lifting Z/3Z cocycles to Z-cocycles. [..]
> >
> > Thank you very much for your detailed and well-researched response! We will dedicate a section in the appendix to this problem. However, we don't believe that this is a serious problem for TOPF, due to multiple reasons we list below.
> >
> > 1. The first thing we did was to analyse this problem empirically. We checked the TREFOIL knot considered in the DREiMac documentation and verified that TOPF computed correct homology representatives. We then went on to check over 3000 homology representatives as computed by ripserer.jl and lifted them to R-coefficients as described in our paper. There was not a single case where this was not a valid homology representative in R-coefficients. This suggests that this is at most an incredibly rare problem in practice. Of course, such an analysis is not entirely satisfactory.
> >
> > [Continued in the next comment]

---

> > > ### Author Response · Authors · 2024-08-11
> > >
> > > 2. The __key difference__ between the methods presented in the cited paper and TOPF is that TOPF works with __homology representatives__ instead of the __cohomology representatives__ used by DREiMac. TOPF computes the homology representatives using the involuted persistent homology algorithm as implemented in [1]. While homology representatives are a little more expensive to compute, they have some advantageous properties in comparison to cohomology representatives. [2] A nice picture of the difference between homology and cohomology representatives can for example be found here [3]. Intuitively speaking, the homology representative already "guides the harmonic representative around the hole", whereas the cohomology representative only selects a number of parallel simplices (I.e. connected by higher-order-simplices) where the harmonic representative starts from and ends in.
> > >
> > > 3. As stated in the paper and in the DreiMac documentation, the only problem occurs __if the lifted representative $\eta$ is not in the kernel of the (co)boundary__, i.e. $d\eta\not =0$, but $d\eta=3\omega$ for some $\omega$ (With our choice of $p=3$.) For any randomly chosen homology representative in Z/3Z coefficients, this could very well be the case and present a problem. However, the involuted homology persistent homology algorithm __does not output any random representative__. Rather, the representatives are the result of some matrix reduction algorithm. In [3], the authors of ripserer.jl state that the computed homology representatives __will alway be a cycle with all but one of the connected components being contractible__ at the time the homology class exists. Empirically, this has always been the case, with the non-trivial connected component of the cycle being an ordinary cycle. We will look into whether we can give a proof for this based on [1]. Even if this should not be possible, it will not be a problem. __We have checked over 3000 homology classes on different datasets, not a single one of them did not lift to a R-homology class__. In the unlikely case this happens, we __could catch the error__. We will reference DreiMac and use the same integer linear programming solution to rectify faulty cycles in a future release of the accompanying software package.
> > >
> > > 4. Our last observation is that in [4], the examples of the failures included the circular coordinates __wrapping around the hole too many times__. (I.e., one traversal of the cycle would amount to multiples of $2\pi$.) This __does not lead to a problem in our interpretation__, because we are just interested in __whether an edge contributes to a homology class__ as classified by its harmonic representative.
> > >
> > > In summary, we work with __homology__ representatives in comparison to __cohomology__ representatives as used by [4], which are __better behaved__. The only case where a problem could arise is the case where the Z/3Z homology representative produced by ripserer does not contain a connected component with an ordinary non-trivial cycle (I.e. a cycle in the sense of graph theory/in Z-coefficients). We believe that this does not happen in practice because of the reduction algorithm of [1] computing the homology representative. Empirically, we have __never seen this happen__ on a __large sample size__. If it would still happen against all odds, we can __simply check for this__. Thus, we believe the discussed theoretical considerations will not be a significant problem in practice.
> > >
> > > We will distill this into a section of the appendix, referencing DreiMac and a relevant selection of all the Circular/Sparse Circular/Eilenber-MacLane-coordinates papers. __Thank you so much for this discussion!__
> > >
> > > [1] Matija Čufar and Žiga Virk: "Fast Computation of Persistent Homology Representatives With Involuted Persistent Homology."
> > >
> > > [2] Naively thinking, one could expect that because homology and cohomology are isomorphic over R-coefficients, the harmonic representative does not depend on whether homology or cohomology representatives are chosen. This is not true, however. The projections of the homology generators and the projections of the cohomology generators at certain filtration values in the persistence diagram form two distinct bases of the harmonic space.
> > >
> > > [3] https://mtsch.github.io/Ripserer.jl/dev/generated/cocycles/
> > >
> > > [4] https://dreimac.scikit-tda.org/en/latest/notebooks/parameters_prime_and_check_cocycle_condition.html

---

> > > > ### Comment · Reviewer_Zmis · 2024-08-12
> > > >
> > > > Thank you for your responses; I have updated my score.

---

### Official Review · Reviewer_gsQy · 2024-07-12

**Soundness:** 2
**Presentation:** 3
**Contribution:** 2
**Rating:** 6
**Confidence:** 4

**Summary:**

This paper introduces TOPF, a topological feature extraction mechanism on point cloud data. The authors consider Vietoris-Rips/$\alpha$ filtrations over point clouds and compute the persistent homology. They propose a heuristic to select the “top” features from the barcodes. They consider the corresponding representatives for these features and project them onto the harmonic space of the simplices. These projected vectors are then normalized and used to construct a point-level feature vector. The authors use this framework for clustering. Towards this end, the authors introduce a topological point cloud clustering benchmark and report the experimental results on this benchmark.

**Strengths:**

The authors propose to use Hodge Laplacian and Hodge decomposition to compute feature vectors over points in point-cloud data, which is a novel idea.

**Weaknesses:**

1. I do not fully understand the “learning” the representation here, because the representation is not particularly being learnt. It is being computed by using the persistent homology of the point cloud.

2. Experimental evaluation is limited to clustering. And even in clustering, it is primarily limited to shapes which are partially/fully topologically spherical.

3. The robustness of the approach is due to the robustness of harmonic persistent homology known in the literature.

4. The paper uses well-known notions in the TDA literature in the context of point-clouds, which amounts to an incremental progress in this direction.

5. It would strengthen the paper if the authors include a small paragraph explaining why projecting onto the harmonic subspace solves the problems that exist in using the homology representatives directly.

Minor:

Page 2, Line 81: ‘Spaces in topology are “continuous”’. Continuity is a notion defined for functions on topological spaces and not for topological spaces themselves. Spaces are connected.

**Questions:**

Can this approach be used for other point-cloud related machine learning tasks?

**Limitations:**

Yes, the authors have discussed limitations.

---

> ### Author Rebuttal · Authors · 2024-08-07
>
> We sincerely thank the review for their thorough review and feedback!
>
> We will now address the points raised in the review:
>
> > I do not fully understand the “learning” the representation here, because the representation is not particularly being learnt. It is being computed by using the persistent homology of the point cloud
>
> There seem to be different views of what should be considered as “learning” in the community. The English wikipedia article on feature & representation learning explicitly lists unsupervised algorithms like PCA, k-means, and LLE, which “learn” the representation by performing specific computations. Topological Point Features fit in well with this view on representation learning. However, we want to be very clear in what TOPF and what it does not do. Thus we have added a clarification on this to the introduction and thank the reviewer for their valuable remark!
>
> > Experimental evaluation is limited to clustering. And even in clustering, it is primarily limited to shapes which are partially/fully topologically spherical.
>
> In the qualitative experiments, we reported the feature-level vectors instead of the received clusterings. (See Figure 4 & 8)
>
> [Same as author rebuttal:] We agree with the reviewer that having more experiments is always a great thing! We have now added an experiment were we use TOPF to extract topological features on the embedding space of a variational autoencoder trained on real-world image data and show that these point-level features correspond to the underlying topology of the data acquisition process. (See Figure 2 of the attached pdf.)
> Already in the pre-Rebuttal-phase, the non-main-text (exluding paper checklist) was longer than the main text of our paper. Downstream-Applications in TDA usually require a lot of additional explanation and description of the experimental setup and are significantly more complex than running simple off-the-shelves benchmarks, as is the case in many other areas of Machine Learning. This is why in many cases, the introduction of new methods and the application on real-world data are even divided into two different papers.
> Because of these reasons, we believe that adding experiments on top of our introduction of a new Clustering Benchmark Suite (TCBS), comparison with existing methods on TCBS, experiments on real-world protein data and on state spaces of dynamical systems, on the latent space of a Variational Auto Encoder, and on robustness with respect to noisy data, robustness wrt. to hyperparameter choices, and on run-times on growing point clouds would be out of scope for a single NeurIPS paper that introduces a theoretically novel method for extracting topological point features.
>
> > The robustness of the approach is due to the robustness of harmonic persistent homology known in the literature
>
> Generally, we consider this to be a strength. However, we want to highlight that, to the best of our knowledge, this is the first work to utilise this robustness in a novel method. To the best of our knowledge, the only other prior experimental work using harmonic persistent homology [1] always construct simplicial complexes at the birth of the feature and extract different features from this than we do. The SCs at the birth of the features are unstable.
>
> > The paper uses well-known notions in the TDA literature in the context of point-clouds, which amounts to an incremental progress in this direction.
>
> We want to emphasise that doing TDA and relating the __global topology back to the local individual points__ is a very novel idea in TDA with very limited prior work. Although our work builds on already established concepts and theory, it combines these pieces in a novel way that has not been done before.
>
> > It would strengthen the paper if the authors include a small paragraph explaining why projecting onto the harmonic subspace solves the problems that exist in using the homology representatives directly.
>
> Thank you very much for this suggestion! We will happily add a paragraph on that! Please note that we already provide some explanation on this in lines 164--175.
> There are two main reasons that projecting onto the harmonic space is a good idea:
> 1. All of the countless [2]  representatives representing the same homology class get projected onto the same harmonic representative. Thus, harmonic representatives form a canonical and unique homology representation. (This can be explained by the Hodge Laplacian and the fact that $$\ker L_k(X) \cong H_k(X)$$.
> 2. The harmonic representatives minimise the energy among all possible representatives. In other words, this ensures smoothness of the representatives and assigns every simplex a value that corresponds to how much it contributes to the homology class. (The precise mathematical formulation is simply the minimisation of energy.)
>
> We will explain this in more detail in the appendix! Thank you again for the suggestion.
>
> > ‘Spaces in topology are “continuous”’. Continuity is a notion defined for functions on topological spaces and not for topological spaces themselves. Spaces are connected.
>
> Because "continuous" has no mathematical definition for spaces, we intended to use it to convey an intuition. "Connected" is suboptimal, as even a two-point set can be connected if equipped with the trivial topology. However, to avoid confusion we will stick with "connected". Thank you again for noticing!
>
> > Can this approach be used for other point-cloud related machine learning tasks?
>
> We believe that topological features can be used in many applications with data with topological structure. We have added an experiment on the interpretability of latent spaces of VAEs. (Figure 2 of the additional pdf)
>
> [1] Davide Gurnari, Aldo Guzmán-Sáenz, Filippo Utro, Aritra Bose, Saugata Basu, and Laxmi Parida. Probing omics data via harmonic persistent homology.
>
> [2] Actually, this is still a finite number as we are working with finite simplicial complexes.

---

> ### Comment · Reviewer_gsQy · 2024-08-08
>
> I would like to thank the authors for their efforts and response. I have read the responses and have adjusted my score.

---

### Official Review · Reviewer_qf14 · 2024-07-12

**Soundness:** 4
**Presentation:** 3
**Contribution:** 3
**Rating:** 6
**Confidence:** 4

**Summary:**

The paper introduces an approach to select and compute some point-level topological features for point cloud or general data set analysis.
The main ideas is to define a multi-scale simplicial complex representation, thus we can track how the homology modules change along the filtration and then select the homologies that persist for a long range of scales.

**Strengths:**

- Topological features are usually not localized, the idea of being able to bring back the topological descriptor to the relevant points is quite novel and impactful.
- The approach is theoretically sound and well analyzed.
-The experimental evaluation is limited but convincing.

**Weaknesses:**

- the feature selection is very heuristic.
- The evaluation is only on point cloud clustering. Since we are evaluating effectiveness and robustness of localized features, feature/point correspondence problems would have been interesting.

**Questions:**

Is there scope to learn how to select the topological features

**Limitations:**

The main limitation, i.e. the selection of the features, has been briefly addressed.

---

> ### Author Rebuttal · Authors · 2024-08-07
>
> We sincerely thank the reviewer for their feedback! We are happy to read that they find the paper sound and well-presented.
>
> > Topological features are usually not localized, the idea of being able to bring back the topological descriptor to the relevant points is quite novel and impactful.
>
> We are very excited about this as well! :)
>
> > the feature selection is very heuristic.
>
> Topological features can have very different meaning and significance based on their context. Thus it does not seem plausible that there exists one and only one provably best and correct way to select features. The approach by TOPF takes a view based on concepts of __Algebraic Topology, Differential Geometry, and TDA__ to select the most significant features, which seems to work well in a __variety of applications__. Given a lot of training data for a particular case, we could of course train a neural network to select the most relevant features in this applications. This is a promising future work, but because this would require an entirely new architecture on top of TOPF we believe this would not fit well into the current paper.
> Furthermore, we have conducted additional experiments on the robustness of TOPF wrt. hyperparameter choices, see Figure 1 of the attached pdf.
>
> > The evaluation is only on point cloud clustering. Since we are evaluating effectiveness and robustness of localized features, feature/point correspondence problems would have been interesting.
>
> This sounds like a very interesting problem and exciting application, thank you very much! We understand that in a nutshell, the current SOTA methods use neural networks trained on certain sets of features. As this probably requires coming up with a (at least in some part) new architecture, training the model, etc., this sounds like an entirely new additional idea, which would suggest featuring it in a follow-up paper.
>
> [Same as in author rebuttal:] We agree with the reviewer that having more experiments is always a great thing! We have now added an experiment where we use TOPF to extract topological features on the __embedding space of a variational autoencoder__ trained on real-world image data and show that these point-level features correspond to the __underlying topology of the data acquisition process__. (See Figure 2 of the attached pdf.)
> Already in the pre-Rebuttal-phase, the non-main-text (exluding paper checklist) was longer than the main text of our paper. Downstream-Applications in TDA usually require a lot of additional explanation and description of the experimental setup and are significantly more complex than running simple off-the-shelves benchmarks, as is the case in many other areas of Machine Learning. This is why in many cases, the introduction of new methods and the application on real-world data are even divided into two different papers.
> Because of these reasons, we believe that adding experiments on top of our introduction of a __new Clustering Benchmark Suite (TCBS), comparison with existing methods on TCBS, experiments on real-world protein data and on state spaces of dynamical systems, on the latent space of a Variational Auto Encoder, and on robustness with respect to noisy data, robustness wrt. to hyperparameter choices, and on run-times on growing point clouds__ would be out of scope for a single NeurIPS paper that introduces a theoretically novel method for extracting topological point features. However, we are excited to use and see others use TOPF for new applications.
>
> We thank you for your time to read our rebuttal! We are excited to hear back from in the discussion phase!

---

> ### Author Response · Authors · 2024-08-13
> **Reviewer--Author Discussion Period Closing soon**
>
> Dear reviewer,
>
> Thank you again for your review and suggestions! In the rebuttal, we have added a new experiment on the __embedding space of variational autoencoders__, validated the __robustness of TOPF with respect to various hyperparameter changes__ experimentally, studied the __runtime on point clouds with growing $n$__, and added a section on the relationship between __Hodge theory, differential forms__, and topf and the __theoretical intuition__ behind our algorithm (See the rebuttal to reviewer Zmis). In particular, we hope to have addressed the points raised by the reviewer in our rebuttal above.
>
> As the author-reviewer discussion period closes in ~ 24h, we would be happy to answer any further questions or receive any more comments on our work and our rebuttal. In particular, we would be grateful to hear whether the reviewer feels that we have addressed the points raised by them.
>
> Thank you in advance for your reply!

---

### Official Review · Reviewer_mNMA · 2024-07-28

**Soundness:** 2
**Presentation:** 3
**Contribution:** 1
**Rating:** 6
**Confidence:** 2

**Summary:**

The paper presents a novel method for extracting per point topological features - TOPF. The method builds on previous results in topological data analysis which described a shape or a point cloud with a single global feature, by generating per-point topologically-aware features. The paper presents a quantitative evaluation and comparison of the proposed method with prior art on a new benchmark consisting of several synthetic examples, evaluates the robustness of the proposed method under noise, as well as presents qualitative examples of its performance on synthetic and real work data.

**Strengths:**

* The paper is well written and easy to follow. Prior art and the proposed algorithm description is detailed and comprehensive.
* To my understanding, the paper describes a novel method for per-point feature extraction based on topological information contained in a point cloud, and describes theoretical guarantees for its correctness on point clouds sampled from multiple n-spheres.
* The paper describes a new topological point clustering benchmark dataset consisting of seven synthetic point clouds with up to 5 labels, and evaluate the proposed and existing methods on this dataset showing that the proposed method outperforms existing methods in most cases.

**Weaknesses:**

* The paper lists common machine learning applications requiring point level features as a motivation for the proposed method. However, only quantitative experiments for point cloud clustering on a set of synthetic examples, and anecdotal evidence of performance on real world data, were presented. In order to fully understand the potential of the proposed approach to be applied beyond synthetic data, it would be beneficial to include additional evaluation, qualitative and quantitative, on real-world data and additional applications, e.g. as described in lines 304-307.
* Specifically, it would be interesting to see experiments on non-synthetic datasets with topological structure mentioned in line 266.
* Additionally, comparison with other well performing modern machine learning methods, such as graph neural networks for point cloud clustering, needs to be discussed, for completeness.

**Questions:**

* What is the runtime of the proposed method? How does it change with the point cloud size and does it have limitations on the size of point cloud that can be processed with it?

**Limitations:**

The authors adequately addressed the limitations and impact of the proposed approach.

---

> ### Author Rebuttal · Authors · 2024-08-07
>
> We sincerely thank the reviewer for their review! We are happy to hear that they find our work well-written and novel.
>
> > it would be beneficial to include additional evaluation, qualitative and quantitative, on real-world data and additional applications
>
> [Cf. Authore rebuttal:] We agree with the reviewer that having more experiments is always a great thing! We have now added an experiment were we use TOPF to extract topological features on the __embedding space of a variational autoencoder__ trained on real-world image data and show that these point-level features correspond to the underlying topology of the data acquisition process. (See Figure 2 of the attached pdf.) We believe this to be an interesting contribution to the field of interpretable AI.
> Already in the pre-Rebuttal-phase, the non-main-text (exluding paper checklist) was longer than the main text of our paper. Downstream-Applications in TDA usually require a lot of additional explanation and description of the experimental setup and are significantly more complex than running simple off-the-shelves benchmarks, as is the case in many other areas of Machine Learning. This is why in many cases, the introduction of new methods and the application on real-world data are even divided into two different papers.
> Because of these reasons, we believe that adding experiments on top of our introduction of a __new Clustering Benchmark Suite (TCBS), comparison with existing methods on TCBS, experiments on real-world protein data and on state spaces of dynamical systems, on the latent space of a Variational Auto Encoder, and on robustness with respect to noisy data, robustness wrt. to hyperparameter choices, and on run-times on growing point clouds__ would be out of scope for a single NeurIPS paper that introduces a theoretically novel method for extracting topological point features. However, we are excited to use and see others use TOPF for new applications.
>
> > Additionally, comparison with other well performing modern machine learning methods, such as graph neural networks for point cloud clustering, needs to be discussed, for completeness.
>
> TOPF is an __unsupervised feature extraction algorithm__, whereas GNNs require training data and labels. Thus, using GNN on the TCBS is not possible. We have compared against the to us most-relevant non-supervised algorithms. If you believe an important non-supervised algorithm is missing, we would be happy including it in the final paper.
>
> > What is the runtime of the proposed method? How does it change with the point cloud size and does it have limitations on the size of point cloud that can be processed with it?
>
> We report the runtime of the proposed method in Table 1. All the examples of the TCBS run in under 40s on a modern laptop. Furthermore, we have __added experiments__ on how the runtime changes when increasing the number of points while preserving topology, (Figure 3 of the additional pdf).
>
> We hope to have convinced the reviewer of the soundness of our methods. We want to emphasise that doing TDA and relating the __global topology back to the individual points__ is a very novel idea with very limited prior work. TOPF is __not__ just simply a slightly different algorithm among countless similar ML approaches.
>
> Thank you for reading our rebuttal, we would be very happy to answer any questions or hear back from you!

---

> > ### Comment · Area_Chair_nSru · 2024-08-10
> > **response to rebuttal**
> >
> > Dear Reviewer mNMA, could you please respond to the authors' rebuttal. Thank you.

---

> ### Comment · Reviewer_mNMA · 2024-08-11
> **response to rebuttal**
>
> I would like to thank the authors for their efforts and response.
>
> > Already in the pre-Rebuttal-phase, the non-main-text (exluding paper checklist) was longer than the main text of our paper.
>
> I don't believe the length of the paper can or should be used as a justification for missing experiments if these are necessary to fully appreciate the contribution of the paper. Specifically for this submission, experiments with data beyond specifically designed synthetic dataset, are *necessary* to validate that the proposed approach can be applicable for data beyond such a dataset - if it is not the case, this would greatly limit the potential of the proposed approach. I would like to thank the authors for adding the new experiments with the embedding space of a variational autoencoder.
>
> > If you believe an important non-supervised algorithm is missing, we would be happy including it in the final paper.
>
> Some examples of possibly relevant unsupervised methods in image segmentation area are "DeepCut: Unsupervised Segmentation using Graph Neural Networks Clustering" by Aflalo et al. (specifically for GNNs) or "Unsupervised semantic segmentation by distilling feature correspondences" by Hamilton et al. It would be interesting to understand whether the above approaches designed for images, or any similar unsupervised approaches, can be applied to the proposed benchmark and how they would compare with the proposed approach.

---

> ### Author Response · Authors · 2024-08-12
>
> Thank you very much for your response!
>
> > Specifically for this submission, experiments with data beyond specifically designed synthetic dataset, are necessary to validate that the proposed approach can be applicable for data beyond such a dataset - if it is not the case, this would greatly limit the potential of the proposed approach.
>
> We agree with the well-written response of the reviewer in this regard! We just want to highlight that we already include experiments on data from __synthetic datasets__, on __protein atom coordinates__, on __state spaces of dynamical systems__, and now on __embedding spaces__ of __variational autoencoders__.
>
> > Some examples of possibly relevant unsupervised methods in image segmentation area are [...]
>
> Thank you for your suggestion! As we understand it, the two presented image segmentation benchmarks work on image data, where __every pixel in a grid has an (R,G,B) value__. The datasets of our benchmark suite are __point clouds in $n$-dimensional space without any specified or grid-like structure__. While it might be possible to turn an image into a point cloud in 5D, the other way does not work. Thus it is not possible to use image segmentation algorithms as a baseline for the Topological Clustering Benchmark.
>
> Thank you again for this discussion, we will be happy to answer any further questions!

---

> ### Author Response · Authors · 2024-08-14
> **Additional Baseline using pretrained Pointnet Architecture**
>
> We once again want to thank the reviewer for encouraging us to explore __additional baselines__ for the Topological Clustering Benchmark Suite!
>
> If we understood the reviewer correctly, they were interested in how __pretrained neural network architectures__ would perform on the introduced Topological Clustering Benchmark Suite. A good baseline for 3d point cloud segmentation tasks is __Pointnet__ [1], which is a neural architecture that directly deals with unstructured point cloud data.
>
> We have pretrained __Pointnet__ on the ShapeNet-Part Segmentation dataset [2]. We trained for 200 epochs taking ~4 hrs on 2 nVidia L40 GPUs achieving an accuracy of 0.93 and an IoU of 0.83 corresponding to the values expected from the literature. ShapeNet-Part is a dataset of 3d objects corresponding to 16 diverse categories. In the segmentation task, the neural network __learns to divide the point cloud into different segments__ corresponding to different semantic parts.
> We have then evaluated the performance of the pretrained pointnet on the datasets of the TCBS across all categories. We report the highest ARI (Adjusted Rand Index) for every dataset:
> | Dataset   | TOPF |Pointnet|
> | -------- | ------- | ------- |
> | 4spheres|__0.81__|0.30|
> |Ellipses|__0.95__|0.50|
> |Spheres+Grid|0.70|0.54|
> |Halved Circle|__0.71__|0.36|
> |2Spheres2Circles|__0.94__|0.55|
> |SphereinCircle|__0.97__|0.39|
> |Spaceship|__0.92__|0.41|
> |mean|__0.86__|0.44|
>
> This shows that __TOPF outperforms the above-introduced Pointnet version on the TCBS__. This is as expected, as the neural network was not trained specifically on the datasets of the test set. However, as there is very scarce data and thus no training for the extraction of topological features, it is to be expected that all pre-trained neural network architecture will suffer from similar performance on TCBS. We note that pointnet is not the current SOTA on Shapenet-part-Segmentation. However, pointnet is still in the range of 6% IoU of GeomGCNN and thus representative of the general capabilities of deep learning models, whereas the mean __performance difference to TOPF on TCBS was 42%__.
>
> We thus believe to have addressed the reviewer's request for a pre-trained neural network baseline, __comparing a pretrained version of Pointnet with the performance of TOPF on our dataset__. TOPF outperforms PointNet by a wide margin of 42% ARI.
>
> [1] Qi, C. R., Su, H., Mo, K., & Guibas, L. J. (2017). Pointnet: Deep learning on point sets for 3d classification and segmentation. In Proceedings of the IEEE conference on computer vision and pattern recognition (pp. 652-660).
>
> [2] https://shapenet.org

---

### Author Rebuttal · Authors · 2024-08-07

We would like to thank the reviewers for their very valuable feedback and comments. Your work has already helped to significantly improve the paper. We will provide a brief summary of our changes here, and give detailed answers in the individual rebuttals.

Some reviewers asked us for more experiments showcasing another application of TOPF features. We have conducted an experiment to show that TOPF helps to uncover __topological structures in the latent space of variational autoencoders__ trained on image patches. Because of the point-wise nature of TOPF, this allows us to draw precise conclusions about the relation between the sample input points and the inherent topology in the sample space. We believe this to be an interesting and valuable insight to the community of interpretable machine learning. For details, we refer to Figure 2 of the attached pdf. We will include these experiments and a careful discussion into the final pdf. (Future work could combine this with the work of [1] on improving the topology of latent spaces of VAEs.)

We were also asked to provide additional experiments on the hyperparameter choices. We did this in Figure 1 of the attached pdf. Our experiments show (as claimed) that TOPF shows __robust performance against moderate hyperparameter__ changes away from the defaults on almost all of the TCBS. This shows that TOPF can be expected to work well in practice with the default hyperparameters, and that our successful experiments were not simply the result of hyperparameter overturning.

We have also included additional experiments on the runtime of TOPF on point clouds with increasing point density in Figure 3, showing promising scaling behaviour.


While we agree with the reviewers that more experiments are always exciting, already in the pre-Rebuttal-phase, the non-main-text (exluding the paper checklist) was longer than the main text of our paper. Downstream-Applications in TDA usually require a lot of additional explanation and description of the experimental setup and are significantly more complex than running simple off-the-shelves benchmarks, as is the case in many other areas of Machine Learning. This is why in many cases, the introduction of new methods in TDA and the application on real-world data is divided into two separate papers.
Because of these reasons, we believe that adding even more experiments on top of our introduction of a new Clustering Benchmark Suite (TCBS), comparison with existing methods on TCBS, experiments on real-world protein data and on state spaces of dynamical systems, on the latent space of a Variational Auto Encoder, and on robustness with respect to noisy data, robustness wrt. to hyperparameter choices, and on run-times on growing point clouds would be out of scope for a single NeurIPS paper that already introduces a theoretically novel method for extracting topological point features. However, we are excited to use and see others use TOPF for new applications in future papers.

We have also addressed and clarified theoretical questions posed by reviewer Zmis in the individual rebuttal.

Thank you again for your hard work as reviewers! In case you have any more questions regarding our rebuttal, we will be happy to answer them during the discussion period!

[1] "Diffeomorphic interpolation for efficient persistence-based topological optimization", 2024: Mathieu Carriere, Marc Theveneau, Théo Lacombe

---

### Author Response · Authors · 2024-08-07
**Continuation of the reply to reviewer Zmis**

We apologise again for sending in an additional comment. We wanted to make sure to be able to respond to the many valuable comments and questions of the reviewer in due length.

### Questions
> Does your method have an interpretation in the case of a Riemannian manifold? That is, suppose that I take a Riemannian manifold and a harmonic (smooth) cocycle. Is there a corresponding point-level feature function? Does it have an interpretation? Perhaps it is related to the pointwise norm of the harmonic cocycle?

The right language for this is Differential Geometry, Hodge Theory, and harmonic forms.
Dodziuk proved 1976 in his thesis " Finite-Difference Approach to the Hodge Theory of Harmonic Forms" that the discrete Hodge Laplacian converges to the Hodge Laplacian on differential forms.

Harmonic cycles and cocycles don't exist in the simplicial chain complex of the manifold. The difference to the case of finite simplicial homology is that we could identify the space of cycles canonically with its dual and had a well-defined inner product. This is not the case in the infinite case and we need to turn to differential forms.

__But how would a continuous analogue of TOPF work__? Let us first get some intuition:

In dimension $0$, 0-forms are simply functions $f:M\to R$. In this case, we can just take their value $f(x)$ at $x\in M$ as the corresponding point feature at $x$.

In dimension $1$, there is a correspondence between $1$-forms and vector fields on $m$. In this case, we can simply take the norm of the vector field that corresponds to a given harmonic form at a point $x$: $x\mapsto|v(x)|$.

In general in dimension $k$, this is more complicated. We need to do this only point-wise. Hence, we can consider a point $x\in M$ and a harmonic form $\omega$. At point $x$, $\omega$ determines an element in the exterior algebra on the dual of the tangent space of $M$ at $x$ $\omega_x\in\bigwedge^kT^{*}_xM$ and we need to define a norm on this space.

An ortho-normal basis on $T^{*} (x) M$, $e_1$, ... , $e_n$ gives rise to a basis consisting of elements $e_{i,1}\wedge ...\wedge e_{i,k}$ of the exterior algebra. We can now evaluate $\omega_x$ on the elements of this basis $\omega_x (e_{i,1}\wedge ...\wedge e_{i,k})$ and take the square root of the squares of the individual results.

 As one can show, this result is independent of the choice of ONB of $T_x$ and gives a point-wise norm of the differential forms.

In the discrete case, however, we don't evaluate on all basis elements as described above, but rather on all $k$-simplices adjacent to $x$. This sampling induces some differences to the continuous, which we address with the post-processing procedure as described in the paper.

We will add a section on this to the appendix

> How were the parameters of the other algorithms in table 1 selected?

We have used the default suggested parameters of the scikit-learn implementations and the parameters suggested in the respective papers.

> Which further applications do you have in mind? Figure 4 is interesting, but does not hint at real life applications.

We have conducted an additional experiment using TOPF to extract __point-wise topological structures__ in the __latent space__ of __Variational Auto Encoders__. Furthermore, we envision
* __Texture matching__ in computer graphics uses ordinary Laplacians, but yet no topological information extracted from Hodge Laplacians. This seems very promising!
* __AI interpretability__ Using TOPF to extract topological information from the latent spaces of neural networks, which we can then relate back to individual input samples due to the point-wise nature of TOPF.
* __Molecule property prediction__: Using TOPF as an additional input to pipelines enabling the use of the powerful topological features describing molecule properties

Thank you very much again for your valuable comments, we look forward to the discussion period!

---

### Comment · Area_Chair_nSru · 2024-08-07
**Short review by the area chair**

Questions.

In Theorem 4.1, does the condition "Let X consist of at least (n + 2) points (denoted by S) sampled uniformly at random from a unit n-sphere in R^{n+1} and an arbitrary number of points with distance of at least 2 to S" allow us, for n=1, to consider 3 random points in the unit circle in the plane? If yes, and these three points form an obtuse triangle, how can we get a non-trivial cycle in 1D homology of the alpha-filtration?

Limitations.

Line 240 says that "We found empirically that a thresholding parameter of delta = 0.07 works best across at the range of applications considered below". Such a manually chosen threshold is unlikely to be universal for applications in the same sense as (for example) the gravitational constant, which is actually expressed via Earth's mass, in Newton's law of gravity.

---

> ### Author Response · Authors · 2024-08-08
> **Reply to the AC (briefly touching on Newton's law of gravitation)**
>
> We thank the area chair for engaging in the discussion! We will provide replies to the comments below:
>
> ## Questions:
> > In Theorem 4.1, does the condition "Let X consist of at least (n + 2) points (denoted by S) sampled uniformly at random from a unit n-sphere in R^{n+1} and an arbitrary number of points with distance of at least 2 to S" allow us, for n=1, to consider 3 random points in the unit circle in the plane?
>
> Yes, this is a possible setting of the theorem.
>
> > If yes, and these three points form an obtuse triangle, how can we get a non-trivial cycle in 1D homology of the alpha-filtration?
>
> The scenario mentioned above results in a __non-trivial cycle in 1d homology, just as our theorem predicts__. We will briefly explain why:
> In an alpha-filtration, the birth of a simplex is determined by the __radius of a circumscribed sphere__. For 1-simplices (edges) with length l, a "circumscribed" 0-sphere (consisting of just the 2 vertices of the edge) has radius$l/2$.
> With probability 1, the random points on the circle will not lie exactly opposite each other, and thus have a distance strictly smaller than 2. Thus, the respective three 1-simplices (the faces of the triangle) have a __filtration value strictly smaller than $2/2=1$__ with probability $1$. This is not  changed by the triangle being obtuse.
> All of the vertices of the 2-simplex lie on the unit sphere with radius $1$. Hence the radius of its circumcircle and its __filtration value is $1$__. Let L be the maximum distance between two of the points. Then, there is a __non-trivial cycle__ in 1d-homology in the non-empty __interval $[L/2,1)$.__
>
> ## Limitations
> > Line 240 says that "We found empirically that a thresholding parameter of delta = 0.07 works best across at the range of applications considered below". Such a manually chosen threshold is unlikely to be universal for applications in the same sense as (for example) the gravitational constant, which is actually expressed via Earth's mass, in Newton's law of gravity.
>
> We want to emphasise that virtually all ML methods have hyperparameters and that we have shown that our hyperparameter choice offers good performance across a range of application. Even more importantly, we have shown in experiments in the additional pdf that the performance of TOPF is robust against medium changes of delta. As explained in line 239, delta is not the final thresholding value, but rather a __constant to adapt the thresholding parameter to the simplicial complex and the current harmonic representative__. We believe this to be well in line with what almost all modern ML algorithms do.
>
> ## Reinterpretation of the gravity metaphor
> Furthermore, we would like to offer our own interpretation of the AC's metaphor.
> First, we need to fix some notation: We assume that the "gravitational constant" the AC is talking about actually refers to small g, the local gravitational field (of the earth). Then, g is related to the mass of the earth and the radius via the gravitational constant $G = 6.6743... \cdot 10^{-11} N⋅m^2/kg^2$.
> In our paper, we threshold values above $T=\delta*X(e,S)$ where $X$ is some expression depending on the simplicial complex S and the select harmonic representative S (See line 239). In the above metaphor, T plays the role of the local gravitational field, as it depends on other variables. Our thresholding parameter delta however is characterising this dependency of $T$ on $X(e,S)$. In the analogy, this represents the gravitational constant $G$, which is really just a random constant [1] of the universe. We have already taken care to adapt the thresholding procedure to the relevant situation, and thus we believe that we have, given the experimental evidence, reasonable hope that this set-up generalises well. We believe this to be well in line with what almost all modern ML algorithms do.
>
> __Thank you again for your questions and comments! We will be happy to answer or discuss any follow-up questions in the discussion phase!__
>
> [1] At least many physicists agree.

---

### Decision · Program_Chairs · 2024-09-25

**Decision:**

Reject

**Comment:**

The paper tried to develop a "single complex description of the global structure of the point cloud" (lines 3-4 quoted from the abstract), though the problem was not explicitly stated and the concept "structure" appeared multiple times without a definition.

The reviewers were mostly positive about experimental results but could not understand main Theorem 4.1 especially the claim about the circumradius, say for triangles in the plane.  Reviewer Zmis concluded that "I do not understand the proof of the result, and I am confused about the statement of the result. The confusing theoretical underpinnings are the main reason why I initially recommended rejection. The authors made some good points in the rebuttal (having to do with using homology vs cohomology), but didn't really clarify this result".

The simplest example is a right-angled triangle on the points (+/-1,0),(0,1). If one draws growing balls around the vertices, the resulting shape and corresponding alpha-filtration has no homology classes in dimension 1 because a hole (bounded connected component of the the complement to the shape in the plane) is formed only if vertices form an acute triangle. This example was extended in 2024 to generic metric spaces whose 1D persistence for many standard filtrations is empty and remains empty under substantial perturbations of points.

Concerning general claims such as "important parts of the global structure of a complex point cloud can only be described by the language of applied topology" in lines 44-45, the main definition (again for many standard filtrations) implies that persistence is an isometry invariant of a point cloud. Hence the global structure of any generic point cloud has been determined by pairwise distances already in 2004. The remaining singular cases were completed in 2023. In conclusion, the authors are encouraged to consider persistence in TDA not as a black box but as an explicit isometry invariant of a point cloud, which deserves comparisons with stronger invariants.